# Causal role for sleep-dependent reactivation of learning-activated sensory ensembles for fear memory consolidation

Brittany C. Clawson [1], Emily J. Pickup[1], Amy Ensing[1], Laura Geneseo [1], James Shaver[1], John Gonzalez-Amoretti [2], Meiling Zhao[1], A. Kane York[3], Femke Roig Kuhn[1,4], Kevin Swift[5], Jessy D. Martinez[1], Lijing Wang [1], Sha Jiang[1] & Sara J. Aton[1✉]

Learning-activated engram neurons play a critical role in memory recall. An untested hypothesis is that these same neurons play an instructive role in offline memory consolidation. Here we show that a visually-cued fear memory is consolidated during post-conditioning sleep in mice. We then use TRAP (targeted recombination in active populations) to genetically label or optogenetically manipulate primary visual cortex (V1) neurons responsive to the visual cue. Following fear conditioning, mice respond to activation of this visual engram population in a manner similar to visual presentation of fear cues. Cue-responsive neurons are selectively reactivated in V1 during post-conditioning sleep. Mimicking visual engram reactivation optogenetically leads to increased representation of the visual cue in V1. Optogenetic inhibition of the engram population during post-conditioning sleep disrupts consolidation of fear memory. We conclude that selective sleep-associated reactivation of learning-activated sensory populations serves as a necessary instructive mechanism for memory consolidation.

[1] Department of Molecular, Cellular, and Developmental Biology, University of Michigan, Ann Arbor, MI, USA. [2] Universidad Ana G. Mendez, Recinto de Gurabo, Gurabo, Puerto Rico. [3] Neuroscience Graduate Program, University of Michigan, Ann Arbor, MI, USA. [4] Center for Neurogenomics and Cognitive Research, VU University, Amsterdam, The Netherlands. [5] Department of Molecular and Integrative Physiology, University of Michigan, Ann Arbor, MI, USA.
✉email: saton@umich.edu

Post-learning sleep improves consolidation of memory on a wide variety of tasks. Data from human subjects[1–4] and animal models[5–8] indicates that post-training sleep may selectively improve performance on tasks involving emotional valence association or abstraction of implicit rules from multi-sensory information. Consistent with this role, sleep appears to facilitate bidirectional functional communication between neo-cortical and subcortical structures, which is thought to play a role in systems memory consolidation[9–17]. In particular, association of multimodal sensory information with contextual information (e.g., emotional valence) may benefit from communication between neocortical and subcortical structures. However, the precise information relayed between different brain structures during sleep states remains unknown. For example, it is unclear how sensory cortical structures engaged during a learning experience could inform association memory storage during subsequent sleep.

Learning experiences during wake clearly influence neural activity patterns during subsequent sleep. For example, hippo-campal place cells activated during environmental exploration in wake show higher firing rates (reactivation)[18] and/or similar sequences of activity (replay)[19–23] during subsequent sleep. This phenomenon has been observed in multiple brain regions, multiple species, and following a wide range of experiences[12,24–29]. Since sleep loss has a disruptive effect on many forms of memory[9], replay and reactivation may play an instructive role in sleep-dependent memory consolidation[9,30]. To test this, prior work has disrupted network-wide activity during specific sleep oscillations[31–34] or disruption of activity in genetically defined cell types across specific phases of sleep[5,11,13,35]—but not the specific neurons activated during learning itself. Recent work has emphasized the essential role of engram neurons in memory recall[36,37]. To date, however, no studies have applied this technology to the question of sleep-dependent memory consolidation. Thus it remains unclear whether engram neurons in various brain regions play an essential role in consolidation of memories following learning.

Here we test the necessity of sleep-specific engram neuron reactivation for memory consolidation. We describe a form of visually cued fear memory in mice, which is encoded by single trial conditioning (pairing presentation of an oriented grating visual stimulus with an aversive foot shock) and dependent on post-conditioning sleep. Post-conditioning, the mice behaviorally discriminated between conditioned and neutral visual cues, leading to a selective fear memory. This discrimination is disrupted by post-conditioning sleep deprivation. Using this paradigm, we take advantage of recently developed genetic tools to selectively manipulate orientation-selective (i.e., cue-activated) primary visual cortex (V1) neurons. We find that these cue-activated visual engram neurons are selectively reactivated during sleep in the hours following visually cued fear conditioning. Optogenetic stimulation of these neurons in awake behaving mice generates a percept of the fear cue, which is sufficient to drive both fear learning and recall. A period of rhythmic optogenetic activation of cue-activated neurons is sufficient to drive functional plasticity—increasing representation of the cue orientation in surrounding V1 neurons—and their optogenetic inhibition reduces cue orientation preference. Finally, we show that selective sleep-targeted inhibition of cue-activated V1 neurons during post-conditioning sleep is sufficient to disrupt consolidation of visually cued fear memory. Based on these findings, we conclude that neurons that are selectively activated in sensory cortical areas during learning play an instructive role in subsequent sleep-dependent memory consolidation. These neurons likely communicate with valence-signaling neurons in other brain regions (e.g., fear response-evoking neurons in the amygdala) in order to form long-lasting cue-driven fear memories. Thus our data also suggest that neocortical engram neurons may play a role in systems memory consolidation, by communicating with neurons in subcortical structures during sleep.

## Results

**Visually cued fear memory consolidation is disrupted by sleep deprivation.** We first tested the role of sleep in consolidating fear memory associated with a specific visual cue. At lights on (i.e., the beginning of the rest phase; Zeitgeber time 0 [ZT0]), wild-type mice underwent visually cued fear conditioning in a novel arena (context A). During conditioning, three 30-s presentations of phase-reversing gratings (of a specific orientation $X°$, shown on 4 light-emitting diode (LED) monitors surrounding the arena) co-terminated with a 2-s foot shock. Mice were then returned to their home cage and either allowed ad lib sleep for the next 12 h or sleep deprived (SD) for 6 h followed by 6 h of ad lib recovery sleep. At ZT12, fear memory for the visual shock cue was assessed in a distinct novel context B. During two separate tests, mice were exposed to gratings of either the same orientation (i.e., shock cue $X°$) or a different orientation (neutral cue $Y°$) (Fig. 1a). As shown in Fig. 1b, mice allowed ad lib sleep showed significantly higher freezing responses during presentation of the shock cue than presentation of the neutral cue (two-way repeated-measures (RM) analysis of variance (ANOVA), effect of sleep condition: $p = 0.043$, effect of cue orientation: $p < 0.0001$, sleep condition × orientation interaction: $p = 0.039$). Overall (pre-cue) freezing to the test chamber (context B) was similar between sleeping and SD mice, as shown in Extended Data Fig. S1. Both sleeping and SD mice discriminated between the shock and neutral cue ($p < 0.0001$

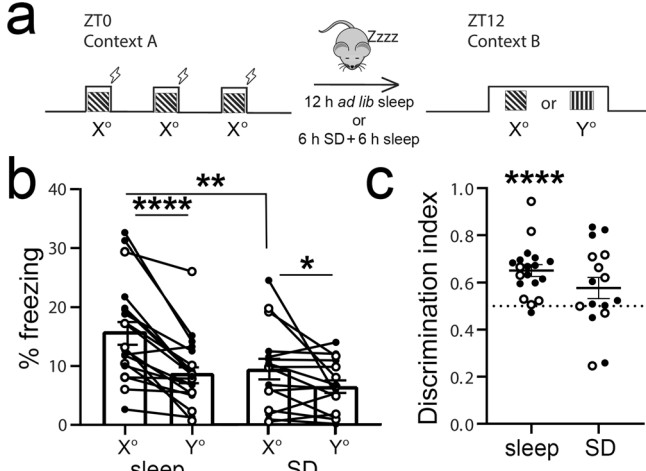

**Fig. 1 Consolidation of visually cued fear memory is enhanced by post-conditioning sleep. a** At ZT0, mice underwent three stimulus–shock pairings in context A. After either 12 h of ad lib sleep or 6 h sleep deprivation (SD) followed by 6 h ad lib sleep, mice were exposed to the shock cue ($X°$ grating) and a neutral cue ($Y°$ grating) in context B. **b** Freezing behavior of the mice during the ZT12 test (Sleep: $n = 19$, SD: $n = 16$; males—solid symbols, females—open symbols). Mice allowed to sleep froze significantly more to the shock cue than mice who were sleep deprived (**$p = 0.007$, Holm–Sidak post hoc test). Both freely sleeping and SD mice showed higher freezing in response to the shock cue (****$p < 0.0001$, *$p = 0.045$, Holm–Sidak post hoc test; two-way RM ANOVA: main effect of sleep condition, $F = 4.448$, $p = 0.043$, main effect of orientation, $F = 27.268$, $p < 0.0001$, sleep × orientation interaction, $F = 4.629$, $p = 0.039$). **c** Freezing behavior quantified a discrimination index [$X°/(X° + Y)$] for each mouse and compared to chance performance (****$p < 0.0001$, Wilcoxon signed-rank test vs. chance). Values in **b**, **c** indicate mean ± SEM.

for sleep, $p = 0.045$ for SD, Holm–Sidak post hoc test); however, SD mice displayed significantly less freezing to the shock cue than mice allowed ad lib sleep ($p = 0.007$, Holm–Sidak post hoc test). To compare discrimination between cues, a discrimination index was calculated. Only freely sleeping mice showed cue discrimination that differed from chance values (Wilcoxon signed-rank test; Sleep: $p < 0.0001$, SD: N.S.). Figure 1c shows data for both female and male mice (males—filled symbols, females—open symbols; a breakdown by sex is provided in Extended Data Fig. S2). Both sexes displayed discrimination between shock and neutral cues when allowed ad lib sleep ($p < 0.001$ and $p = 0.007$ for males and females respectively, Holm–Sidak post hoc test) and impairment when SD (N.S. for shock vs. neutral, Holm–Sidak post hoc test). Both sexes showed significant discrimination from random chance only when sleep was allowed ($p = 0.001$ for male freely sleeping mice and $p = 0.0156$ female freely sleeping mice, N.S. for male and female SD mice, Wilcoxon signed-rank test). Thus, for subsequent analysis, both sexes were used. Together these data suggest that visually cued fear memory consolidation is sleep dependent.

**Targeted recombination in activated populations (TRAP) targets orientation-selective, fear cue-activated V1 neurons.** To characterize and manipulate activity in V1 neuronal populations activated by oriented grating cues (i.e., putative visual engram neurons), we used previously described techniques for TRAP[38]. cfos-CRE$^{ER}$ mice were crossed to mice expressing tdTomato in a cre-dependent manner (*cfos::tdTom*). The mice were presented with either an oriented grating ($X°$) or a dark screen stimulus for a 30-min period (Fig. 2a). Immediately following this presentation, mice were administered tamoxifen and housed in complete darkness for the next 3 days (to prevent additional visually driven recombination in V1). V1 tdTomato expression (quantified 11 days following tamoxifen administration) was significantly higher in mice exposed to gratings; dark screen presentation induced very low levels of V1 expression (Fig. 2b, c; nested $t$ test, $p = 0.0001$). Based on the estimated density of neurons in mouse V1[39], estimated tdTomato-labeling density following $X°$ presentation was between 0.9 and 2.8%.

To test the orientation selectivity of $X°$-activated TRAPed neurons, mice were presented with either the same oriented grating ($X°$) or an alternate oriented grating ($Y°$) prior to sacrifice (Fig. 2d). TRAPed V1 neurons show a significantly higher percentage of cFos expression following re-exposure to the same orientation than following exposure to a different orientation ($X°$—$32 \pm 3\%$ vs. $Y°$—$21 \pm 2\%$; $p = 0.009$, nested $t$ test; Fig. 2e, f). This level and specificity of cFos overlap is comparable to that reported for auditory stimuli in cochlear nuclei (Guenthner et al.). The difference in overlap between mice presented with $X°$ or $Y°$ was not attributable to either the number of tdTomato+ neurons or cFos+ neurons, which were similar between groups (N.S. for both measures, nested $t$ test; Fig. 2g, h). Together these data suggest that TRAP provides genetic access to orientation-selective V1 neurons, which are activated by oriented grating stimuli.

**Optogenetic activation of TRAPed V1 neurons generates an orientation-specific percept.** To further test the cue selectivity of recombination in neurons activated by an oriented grating ($X°$) and to test the behavioral significance of activity in this neuronal population, we expressed ChR2 in $X°$-activated TRAPed neurons (*cfos::ChR2*). As shown in Fig. 3a, *cfos::ChR2* mice implanted with bilateral V1 optic fibers were presented with a single oriented grating ($X°$) for TRAP as described above. A subset of these mice received tamoxifen to induce recombination in activated neurons; a second control cohort was treated and handled identically but

received vehicle with no tamoxifen (see "Methods"). Eleven days later, one of the two variants of the visually cued fear conditioning was performed. The first subset of mice were conditioned at ZT0 using rhythmic (1 Hz) optogenetic activation of TRAPed V1 neurons (rather than oriented grating presentation) as a cue for foot shock (Fig. 3b). Mice were returned to their home cages, allowed ad lib sleep, and tested at ZT12 in a dissimilar context. At this point, mice were presented with oriented gratings of the same orientation used for TRAP ($X°$) and a different ($Y°$) orientation. Presentation of $X°$ elicited significantly greater freezing responses than presentation of $Y°$ in mice administered tamoxifen to induce ChR2 expression (ratio paired $t$ test, $p = 0.008$; Wilcoxon signed-rank test vs. chance, $p = 0.02$) (Fig. 3c). In contrast, control mice given no tamoxifen showed no discrimination between $X°$ and $Y°$ (ratio paired $t$ test, N.S.; Fig. 3d). This is consistent with recent findings in the olfactory system, in which optogenetic stimulation of a specific glomerulus during aversion conditioning generates an odor-specific aversion[40].

In a second set of experiments (Fig. 3e), mice underwent visually cued fear conditioning at ZT0, using $X°$ gratings as the shock cue. At ZT12, they were placed in a dissimilar context, where after a delay they received bilateral 1 Hz optogenetic stimulation of TRAPed V1 neurons. Mice administered tamoxifen to induce recombination in V1 showed significantly greater freezing behavior during optogenetic stimulation than before and after stimulation (Fig. 3f; $p = 0.003$ for each Holm–Sidak post hoc test). In contrast, control mice given no tamoxifen showed no elevated freezing response during light delivery to V1 (Fig. 3g; N.S., RM ANOVA). Both of these results indicate that optogenetic activation of the $X°$-activated TRAPed V1 ensemble is sufficient to generate a percept of the $X°$ visual cue, consistent with the recent data[41]. Moreover, these data demonstrate that optogenetically activated V1 neurons can substitute behaviorally as a visual cue for either encoding or recalling fear memory. Together, this suggests that activity of the $X°$-activated TRAPed ensemble in V1 constitutes an engram for the visual cue.

**Orientation-selective V1 ensembles are reactivated during post cued fear conditioning sleep.** Since sleep facilitates consolidation of visually cued fear memory and the TRAPed ensemble provides cue-selective information, we next evaluated whether the TRAPed population is selectively activated during post-conditioning sleep. We again expressed tdTomato in TRAPed $X°$-activated neurons (*cfos::tdTom*). As shown in Fig. 4a, these mice were presented with $X°$ to induce tdTomato expression and, 11 days later, were cue conditioned using either the same $X°$ oriented grating stimulus or a dissimilar $Y°$ stimulus. They were then returned to their home cage and allowed ad lib sleep over the next 4.5 h, at which point they were sacrificed for V1 cFos immunostaining. When $X°$ was used as the fear conditioning cue, $33 \pm 2\%$ of tdTomato-expressing V1 neurons showed the expression of cFos after subsequent sleep (Fig. 4b)—a level similar to that seen after same orientation grating exposure (Fig. 2e). When mice were instead conditioned using $Y°$ as the shock cue, the percentage of overlap was significantly lower ($26 \pm 1\%$; Fig. 4c). This difference in overlap was not driven by differences in the densities of tdTomato+ or cFos+ neurons, which were similar between $X°$- and $Y°$-conditioned mice (Fig. 4d, e). These data suggest V1 neurons activated by a visually cued learning experience are more likely to remain active during post-learning sleep, consistent with observations of ensemble reactivation in V1 following other types of learning[24]. Thus sleep-associated V1 ensemble reactivation could serve as a plausible substrate underlying visually cued fear memory consolidation.

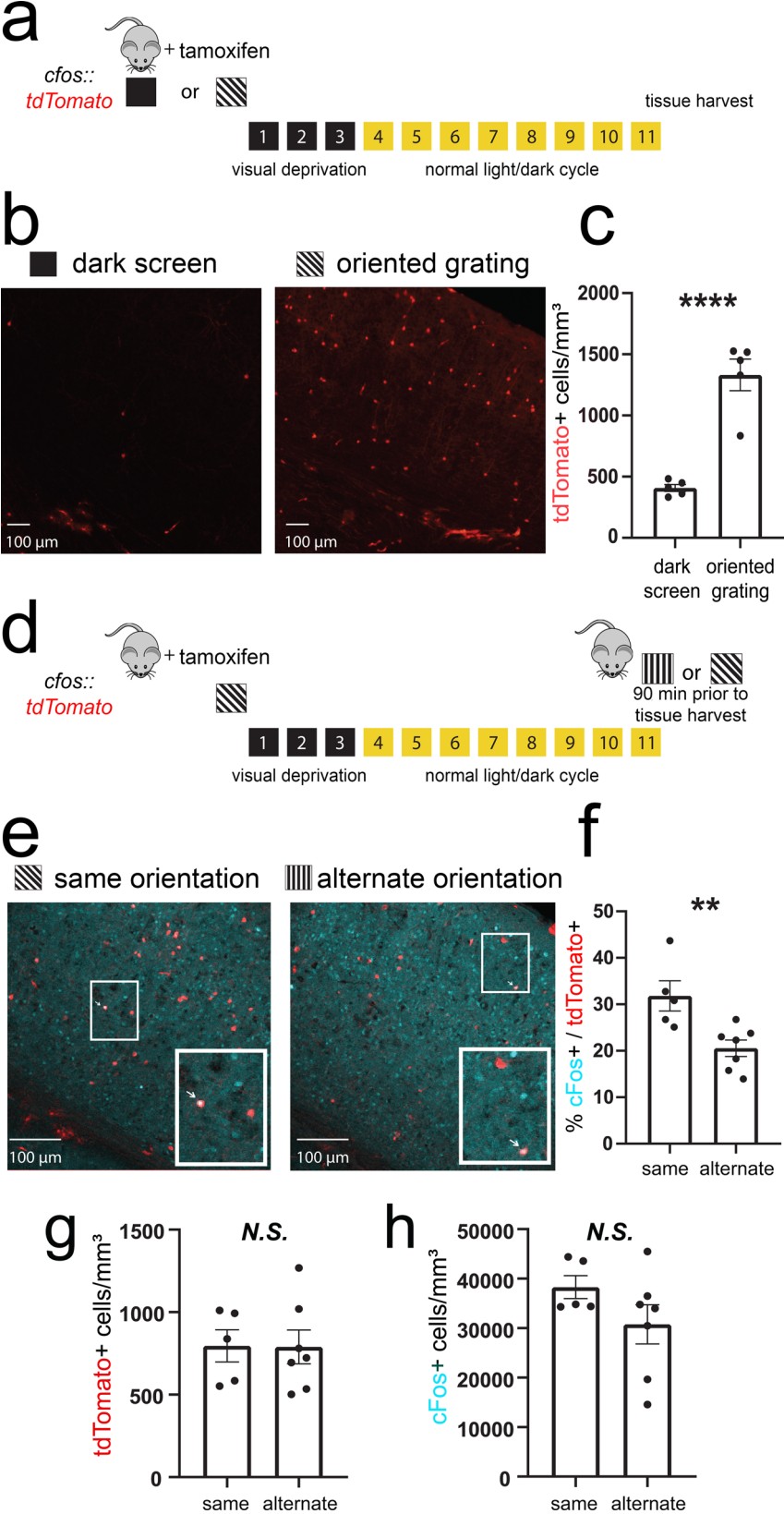

**Rhythmic offline reactivation of orientation-selective V1 ensembles induces plasticity and alters representation of orientation in V1**. To test whether sleep-associated reactivation of orientation-selective neurons could impact the representation of orientation across V1, we tested how rhythmic optogenetic activation of X°-activated TRAPed neurons affected surrounding

V1 neurons' response properties. As described for experiments outlined in Fig. 3, we also recorded from a second control cohort of mice that underwent identical procedures but did not receive tamoxifen following X° stimulus presentation (Fig. 5a). We recorded neuronal firing patterns and visual responses in V1 from anesthetized *cfos::ChR2* mice before, during, and after a period of

**Fig. 2 TRAP labels orientation-selective V1 ensembles. a** *cfos::tdTom* mice were presented with either a dark screen or an oriented grating ($X°$) and were then injected with tamoxifen prior to 3 days of housing in complete darkness. **b**, **c** Representative V1 tdTomato labeling quantified 11 days after tamoxifen administration. ****$p = 0.0001$ ($t = 7.07$, DF $= 8$) for dark screen vs. $X°$, nested $t$ test ($n = 5$ mice/condition) Values in **c** indicate mean ± SEM. **d** Prior to tissue harvest, mice were either re-exposed to gratings of the same orientation ($X°$) or an alternate orientation ($Y°$). **e**, **f** Representative images showing overlap of tdTomato (red) and cFos protein (cyan). An example of colocalization within a neuron (quantified in **f**) is indicated with a white arrow for each image in the inset. **$p = 0.009$ ($t = 3.22$, DF $= 10$), nested $t$ test ($n = 5$ mice for $X°$, $n = 7$ mice for $Y°$). Values in **f** indicate mean ± SEM. **g**, **h** Densities of tdTomato+ and cFos+ neurons were similar in $X°$- and $Y°$-exposed mice. Values indicate mean ± SEM.

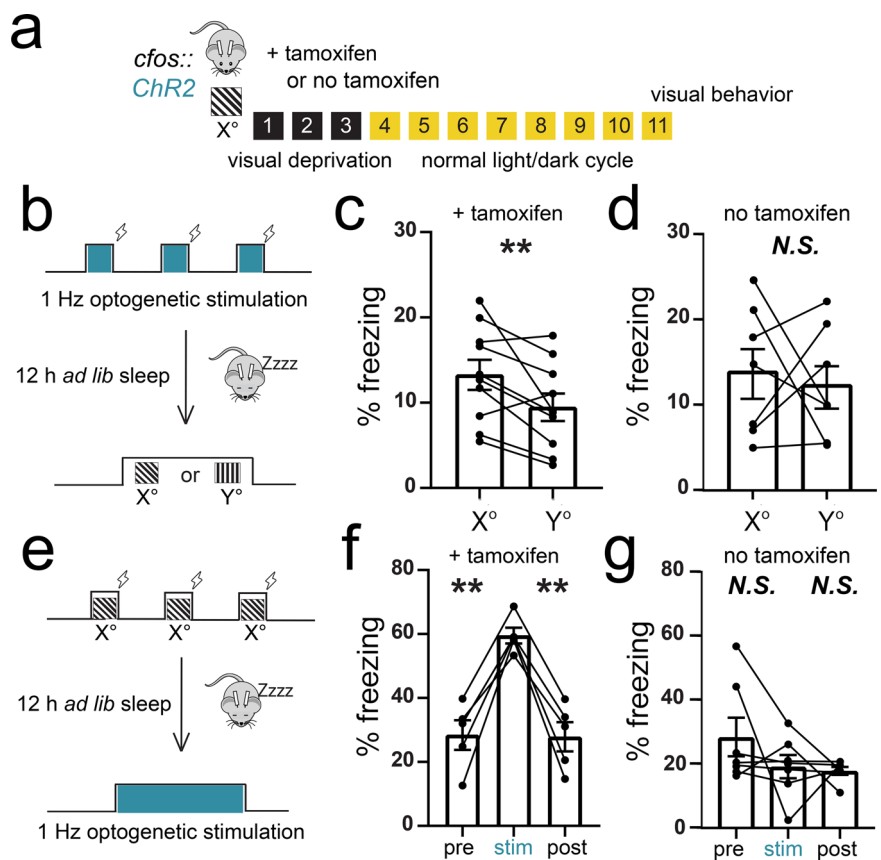

**Fig. 3 Optogenetic stimulation of TRAPed V1 neurons mimics visual experience. a** *cfos::ChR2* mice with bilateral V1 fiber optics had recombination induced to a specific angle ($X°$). As a negative control, a second cohort was treated identically, without tamoxifen administration to induce recombination (no tamoxifen). Eleven days later, visually driven fear behavior was assessed. **b–d** At ZT0, the mice received bilateral V1 optogenetic stimulation paired with foot shocks in lieu of the oriented grating visual stimuli used for cued conditioning in Fig. 1. At ZT12, the mice were presented with the same oriented grating used for TRAP ($X°$) and an alternate orientation ($Y°$). Optogenetically cued conditioning in tamoxifen-administered mice resulted in higher subsequent cued freezing responses to $X°$ relative to $Y°$ ($n = 10$ mice; $p = 0.008$ [$t = 3.38$, DF $= 9$], ratio paired $t$ test). No tamoxifen controls showed no discrimination between $X°$ and $Y°$ (N.S., ratio paired $t$ test). Values in **c**, **d** indicate mean ± SEM. **e–g** At ZT0, a second cohort of mice underwent visually cued fear conditioning to the same orientation as the TRAPed ensemble. At ZT12, the mice received optogenetic stimulation in place of the visual cue. Freezing behavior was higher during the 3-min optogenetic stimulation than before or after stimulation in tamoxifen-administered mice (3 and 1 min, respectively; $n = 5$ mice; pre vs. stim—$p = 0.003$ [$t = 7.30$, DF $= 4$, stim vs. post—$p = 0.003$ [$t = 7.85$, DF $= 4$], Holm–Sidak post hoc test, one-way RM ANOVA). No tamoxifen controls showed no fear response to V1 light delivery (N.S., one-way RM ANOVA). Values in **f**, **g** indicate mean ± SEM.

rhythmic (1 Hz) light delivery. We first generated tuning curves to assess orientation preference for each V1 neuron, measuring firing rate responses to a series of eight different oriented gratings. This orientation preference test was followed by a 20–30-min period without optogenetic stimulation, a second orientation preference test, a 20–30-min period of 1 Hz optogenetic stimulation, and then a final orientation preference assessment (Fig. 5a).

V1 neurons showed heterogeneous firing responses during rhythmic optogenetic stimulation (Fig. 5b and Extended Data Fig. S4). A small fraction of the recorded neurons (4%) were activated immediately following initiation of the 10-ms light

pulses, suggesting direct optogenetic activation by light (Fig. 5b, c). This proportion is very similar to the approximate percentage of neurons labeled with tdTomato by TRAP (Fig. 2). An additional 1% were significantly inhibited, and 1% showed only long-latency (>200 ms) excitatory responses. The remaining recorded neurons were either unaffected by optogenetic stimulation (45%) or showed consistent activation 14–50 ms after light pulses (49%), suggesting that these neurons receive excitatory input from the optogenetically stimulated population (Fig. 5c, left; Extended Data Fig. S4). Rhythmic activation of the $X°$-activated V1 population did not significantly alter the V1 local field potential (LFP) power spectrum (Fig. 5d, N.S., Kolmogorov–Smirnov (K-S) test).

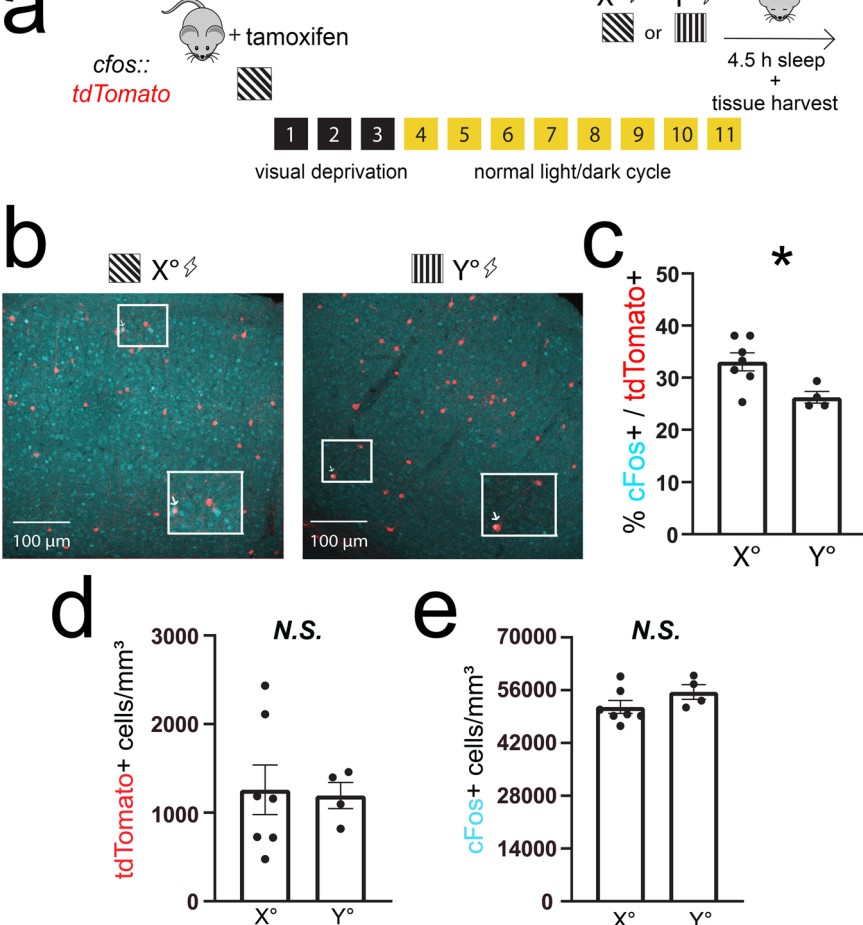

**Fig. 4 TRAPed V1 neurons selective for the conditioned stimulus are reactivated in post-conditioning sleep. a** *cfos::tdTom* mice had recombination induced to a specific angle (*X*°). Eleven days later, they were cue conditioned to either the same angle as induction (*X*°; *n* = 7 mice) or an alternate angle (*Y* °; *n* = 4 mice). All mice were allowed 4.5 h of post-conditioning ad lib sleep prior to tissue harvest. **b, c** Representative images showing overlap of cFos expression (cyan) with tdTomato (red). The boxed region is magnified as an inset with an arrow indicating an overlapping neuron. Expression of cFos in tdTomato-labeled cells was greater for mice conditioned to the same orientation used for TRAP labeling (*\*p* = 0.0106 [*t* = 2.633, DF = 63], nested *t* test). **d, e** Densities of tdTomato+ and cFos+ neurons were similar in *X*° and *Y*°-conditioned mice. Values indicate mean ± SEM.

Rhythmic light delivery had no effect on the firing of neurons in mice that were not administered tamoxifen to induce ChR2 expression in *X*°-activated neurons (Fig. 5c, right).

To assess how optogenetic reactivation of the *X*°-activated TRAPed population affects response properties in surrounding V1 neurons, orientation tuning curves for well-isolated and stably recorded neurons were compared before vs. after optogenetic stimulation. While orientation preference for *X*° (vs. *X* + 90°) did not increase across the 20–30-min period without optogenetic stimulation, a similar period of 1 Hz light delivery caused a selective shift in orientation preference across V1 toward the orientation of the TRAPed population. Shifts in orientation preference toward the orientation of the TRAPed ensemble (*X*°) were statistically significant for those neurons that showed consistent excitatory responses 20–50 ms following light pulses but not for neurons that did not show these responses (N.S. for non-activated neurons, vs. *p* = 0.002 for activated neurons, nested *t* test; Fig. 5e, f). Critically, this shift in orientation preference is similar in nature (if slightly larger in magnitude) to that seen in V1 after a single presentation of oriented gratings, followed by a subsequent period of ad lib sleep[11,42,43]. In neurons recorded from no tamoxifen control mice, no shift in orientation preference was apparent (Fig. 5g).

**Sleep-associated reactivation of orientation-selective V1 neurons is necessary for consolidation of visually cued fear memory.** Because reactivation of orientation-selective V1 populations occurs during post-visually cued conditioning sleep and is sufficient to induce changes in orientation representations in V1, we next tested the necessity of sleep-associated ensemble reactivation for consolidation of visually cued fear memory. To assess how inhibition of the *X*°-activated TRAPed population affects firing in surrounding V1 neurons, we expressed ArchT in cfos-CRE^ER mice (*cfos::Arch*) by administering tamoxifen immediately following presentation of an *X*° stimulus. A second control cohort of mice underwent identical procedures but did not receive tamoxifen (Fig. 6a). We recorded spontaneous activity and visual responses in V1 neurons in anesthetized mice before and during a period of optogenetic inhibition (Fig. 6a). Periodic inhibition (cycles of 5-s light delivery, followed by a 0.5-s ramp off, and 1-s off) led to heterogeneous changes in spontaneous firing (Fig. 6b–d), with 34% showing no response (±0–5% change in firing rate), 21% activated (>5% increase in firing rate), and 45% inhibited (>5% decrease in firing rate). Neurons recorded with identical light delivery in control (no tamoxifen) mice, in comparison, showed a similar proportion of activation (21% showing a >5% increase

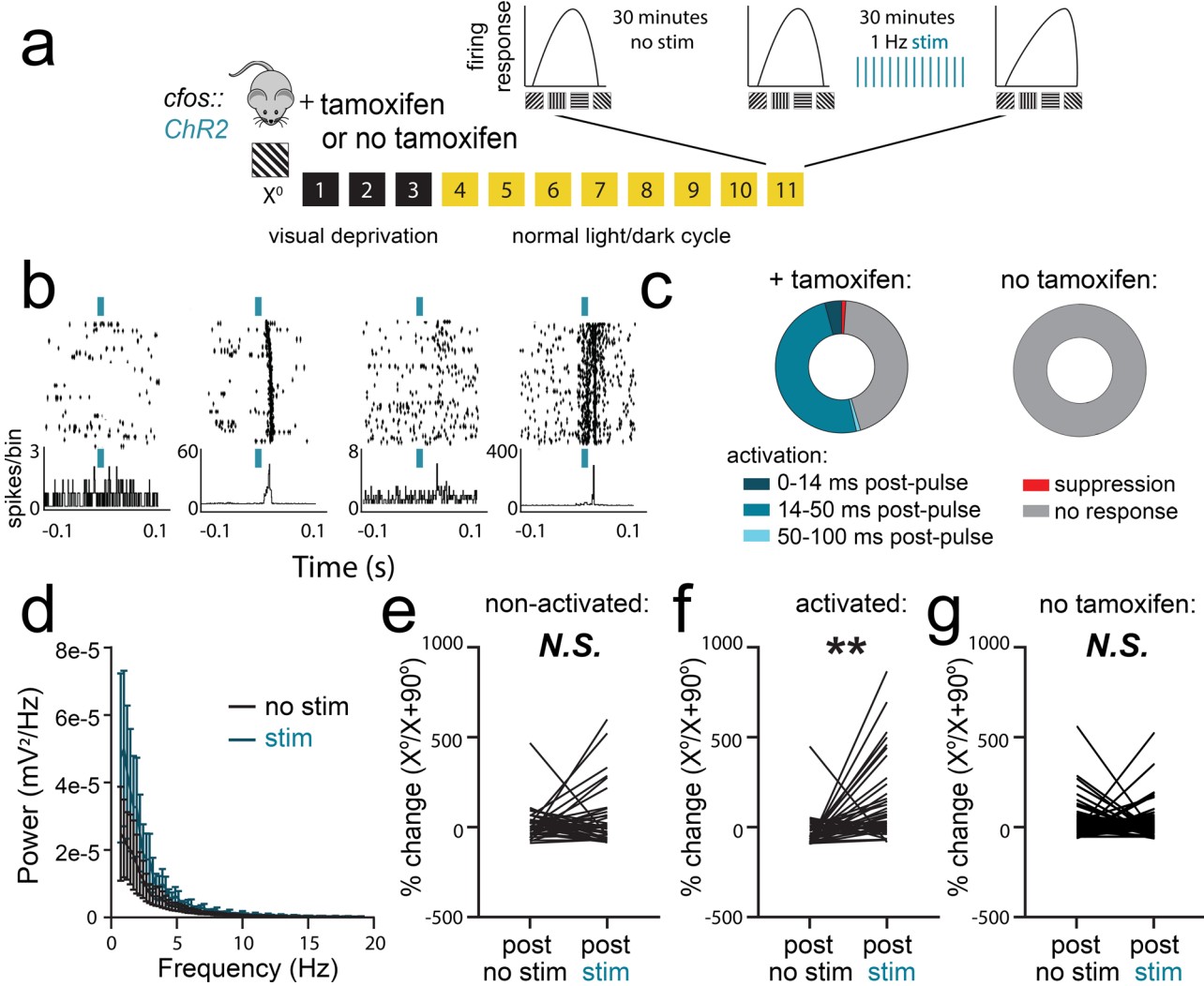

**Fig. 5 Offline reactivation of orientation-selective TRAPed V1 neurons alters orientation representations in V1. a** *cfos::ChR2* mice were presented with an oriented grating ($X°$) for TRAP. A second cohort was treated identically, without tamoxifen administration to induce recombination (no tamoxifen). Eleven days later, orientation tuning was measured repeatedly for V1 neurons recorded from anesthetized mice: at baseline, after a 20–30-min period without optogenetic stimulation, and after a 20–30-min period with 1 Hz light delivery. **b** Representative rasters and perievent histograms for four simultaneously recorded neurons, showing diverse firing responses during optogenetic stimulation of ChR2-expressing neurons. **c** The majority of stably recorded V1 neurons were reliably activated following light pulses, with variable lag times. A small proportion were inhibited by light delivery, and the remaining neurons were not affected ($n = 96$ neurons from 6 mice, total). Neurons recorded during rhythmic light delivery in control (no tamoxifen) mice showed no responses to light pulses ($n = 79$ neurons from 5 mice, total). **d** Power spectra for V1 LFPs showed no significant effect on ongoing rhythmic activity (N.S., K–S test, $n = 5$ mice). Values indicate mean ± SEM. **e, f** After optogenetic stimulation, neurons that were not activated following light pulses showed no change in orientation preference (N.S., nested $t$ test, $n = 37$ neurons from 5 mice). In contrast, activated neurons showed increased firing rate responses for gratings of the same orientation ($X°$) used for TRAP. (**$p = 0.004$ [$t = 3.93$, DF = 8], nested $t$ test, $n = 40$ neurons from 5 mice). **g** Neurons in control (no tamoxifen) mice showed no consistent orientation preference changes following rhythmic light delivery (N.S., nested $t$ test, $n = 79$ neurons from 5 mice).

in firing), less inhibition (15% showing a >5% decrease in firing rate), and a large proportion (64%) not affected. Inhibition did not affect V1 LFP power spectra (Fig. 6e, N.S., K-S test). Light delivery during presentation of oriented gratings led to a significant decrease in orientation preference for $X°$ in inhibited neurons ($p = 0.007$, nested $t$ test) but not in non-inhibited neurons (N.S., nested $t$ test, Fig. 6f, g). Neurons recorded from no tamoxifen control mice showed no significant change in $X°$ preference when light was delivered during presentation of gratings (N.S., nested $t$ test, Fig. 6h). Together, these data indicate that inhibition of the TRAPed ensemble leads to changes in orientation representation across the population, without grossly disrupting network activity across V1.

We next asked whether sleep-targeted inhibition of V1 visual engram neurons (i.e., those encoding the fear memory cues) disrupts consolidation of visually cued fear memory. For these experiments, *cfos::Arch* mice expressing ArchT in $X°$-activated neurons (and control mice not expressing ArchT) underwent visually cued fear conditioning in context A at ZT0, using either $X°$ or $Y°$ as a cue for foot shock (Fig. 7a). They were then returned to their home cages for ad lib sleep. For the first 6 h following conditioning (a window of time where SD disrupts consolidation; Fig. 1), TRAPed neurons in V1 were optogenetically inhibited (using the parameters described for Fig. 6 above) during bouts of non-rapid eye movement (NREM) and rapid eye movement (REM) sleep (Extended Data Fig. S5). This pattern of inhibition

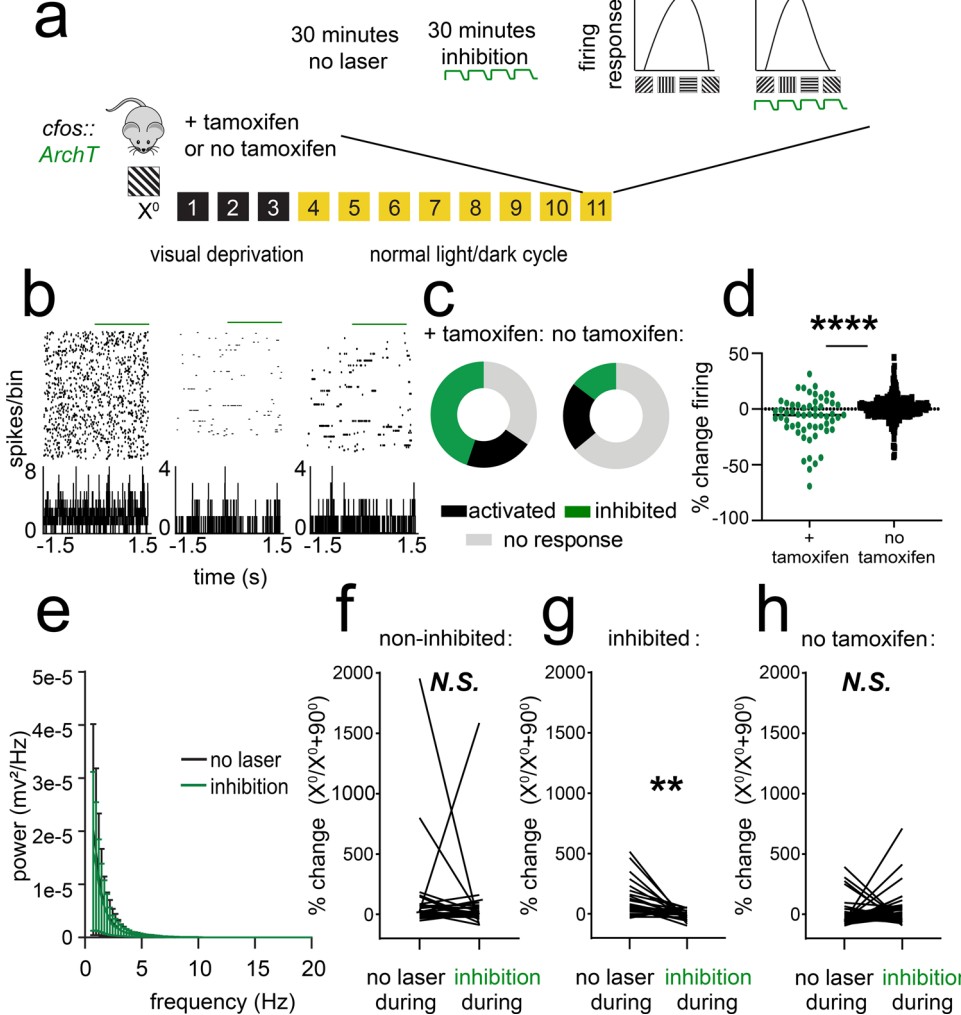

**Fig. 6 Optogenetic inhibition of orientation-selective TRAPed V1 ensembles alters orientation preference in surrounding V1 neurons. a** *cfos::ArchT* mice were presented with an oriented grating (X°) for TRAP. Eleven days later, V1 neurons were recorded from anesthetized mice across 30 min of optogenetic inhibition and 30 min without inhibition. Afterward, orientation preference was assessed at baseline, during a control period without optogenetic inhibition (no laser), and during a period with inhibition delivered at the same time as visual stimuli. A second cohort was treated identically, without tamoxifen administration to induce recombination (no tamoxifen). **b** Representative rasters and perievent histograms for 3 simultaneously recorded neurons from an Arch-expressing mouse, showing diverse firing responses during periodic optogenetic inhibition. **c** Distributions of stably recorded V1 neurons, which were inhibited (with >5% decrease in firing rate), activated (with >5% increase in firing rate), or unaffected by light delivery (*n* = 58 neurons from 5 tamoxifen treated mice, *n* = 278 neurons from 4 no tamoxifen control mice). **d** Firing rate changes with light delivery were significantly greater for Arch-expressing mice than for no tamoxifen control mice (****p* = 0.0001, Mann–Whitney test). **e** Power spectra for V1 LFPs showed no significant change in rhythmic activity during periods of inhibition (N.S., K–S test, *n* = 5 mice). Values indicate mean ± SEM. **f, g** Neurons recorded from Arch-expressing mice that showed no decrease in firing rate during light delivery showed no change in orientation preference when light was delivered to V1 during presentation of visual stimuli (N.S., nested *t* test, *n* = 32 neurons from 5 mice). In contrast, neurons that were inhibited showed a reduced preference for gratings of the same orientation (X°) used for TRAP (***p* = 0.007 [*t* = 3.65, DF = 8, nested *t* test, *n* = 26 neurons from 5 mice). **h** Neurons recorded from no tamoxifen control mice showed no consistent change in orientation preference when light was delivered to V1 during presentation of visual stimuli (N.S., nested *t* test, *n* = 52 neurons from 3 mice).

did not significantly alter either sleep architecture or V1 electroencephalogram (EEG) power spectra (which were similar between inhibited and control mice; Extended Data Fig. S6).

At ZT12, mice were presented with X° and Y° oriented gratings (shock and neutral cues) in a dissimilar context B. In mice presented with X° as a cue for foot shock, sleep-targeted optogenetic inhibition of TRAPed V1 neurons prevented fear discrimination between X° and Y° cues during testing. These mice showed high levels of generalized fear (i.e., high levels of freezing during presentation both X° and Y° gratings and during pre-cue periods; Extended Data Fig. S7), indicating disrupted fear memory consolidation. In contrast, both control mice (not

expressing ArchT) and mice presented with Y° as the shock cue showed cued fear memory consolidation and discriminated between shock and neutral cues at ZT12 (Fig. 7b, c). Together these data suggest that selective reactivation of V1 visual engram neurons during post-learning sleep provides an essential substrate for consolidation of an associative visually cued memory.

## Discussion
Our present data demonstrate that orientation-selective V1 neurons involved in encoding a specific visually cued fear memory (visual engram neurons) play an ongoing role in memory consolidation during subsequent sleep. After selective

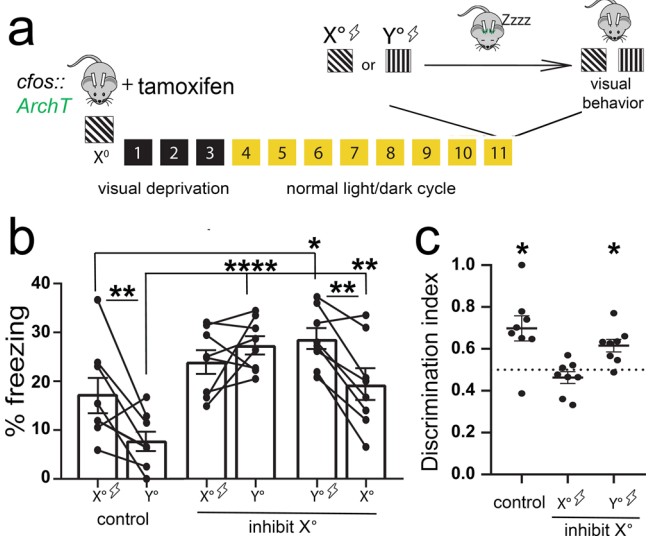

**Fig. 7 Sleep-specific inhibition of a V1 engram disrupts visually cued fear memory consolidation. a** *cfos::ArchT* mice implanted with bilateral V1 optical fibers and EEG/EMG electrodes were presented with *X°* for TRAP. Eleven days later, mice were conditioned using either the same orientation (*X°*) or an alternate orientation (*Y°*) as the shock cue. Post-conditioning, the mice slept with sleep-specific inhibition during the first 6 h. **b** No-inhibition (non-opsin-expressing) controls ($n = 8$) and mice cued to *Y°* with subsequent optogenetic inhibition ($n = 8$) showed higher freezing responses to the shock cue vs. the neutral cue (two-way RM ANOVA: main effect of optogenetic manipulation condition, $F = 10.247$, $p < 0.001$, main effect of orientation, $F = 10.679$, $p = 0.004$, optogenetic condition × orientation interaction, $F = 7.359$, $p = 0.004$, no-inhibition control—$p = 0.002$, *Y°*-cued inhibition—$p = 0.003$, Holm–Sidak post hoc test). In contrast, mice cued to *X°* with subsequent optogenetic inhibition ($n = 8$) did not differ in freezing responses to the shock cue vs. the neutral cue (N. S., Holm–Sidak post hoc test). Mice cued to either *X°* or *Y°* with subsequent inhibition showed higher freezing responses to both cues relative to no-inhibition controls, indicative of generalization. **c** Controls and mice cued to *Y°* show significant discrimination, while mice cued to *X°* did not (*$p = 0.016$ for both no-inhibition control and *Y°* cued with inhibition; Wilcoxon signed-rank test). Values in **b**, **c** indicate mean ± SEM.

activation of these neurons during visually cued fear conditioning, these neurons continue to be active during sleep in the subsequent hours (Fig. 4)—a time window during which sleep plays a role in promoting consolidation (Fig. 1). Activity in these neurons is sufficient to drive a percept that can substitute for the visual fear cue in mice during wake (Fig. 3). It remains unclear how selective sleep-associated reactivation of these neurons affects the surrounding visual cortex (or interacts with circuitry engaged selectively by aversive conditioning). However, periodic optogenetic activation of these orientation-selective neurons is sufficient to drive shifts in orientation preference in surrounding neurons that show excitatory postsynaptic responses to their input (Fig. 5). This leads to an increase in the representation of the visual engram neurons' preferred orientation in the surrounding V1 network. It remains unclear how the magnitude of the changes we observe following selective optogenetic reactivation in anesthetized mice would compare with those caused by spontaneous reactivation in the sleeping neocortex. Moreover, the functions of such an increase in representation are currently unknown. Nonetheless, a similar increase in representation for a specific orientation is seen in the visual cortex in mice[11,42–45], human subjects[46,47], and nonhuman primates[48,49] as a result of orientation-specific sensory experience and task training. Thus

changes in representation in sensory cortex appear to be either a correlate or a cause of changes in orientation discrimination ability with experience.

We show conversely that optogenetic inhibition of orientation-selective neurons acutely reduces the representation for the visual engram neurons' preferred orientation in the surrounding V1 network (Fig. 6). Finally, we demonstrate that optogenetic inhibition of these visual engram neurons during post-conditioning sleep dramatically disrupts consolidation of fear memories for specific visual cues (Fig. 7). Mice with sleep-targeted inhibition of cue-activated neurons show high levels of general freezing behavior at testing but no discrimination between cues of different orientations. Thus their specific memory deficit seems to be due to an inability to link fear memory to a specific orientation cue during consolidation, rather than a disruption of fear memory per se.

This work links together two bodies of literature regarding the neural substrates of memory. One recent area of investigation has focused on the role of engram neurons that are activated by learning experiences and whose activation is necessary and sufficient for memory recall[36,37,50]. However, the role these neurons play in the consolidation of memories following learning has been a matter of speculation. Here we show that the neurons engaged during learning play a necessary and instructive role during subsequent sleep. The second body of literature has focused on replay of learning-associated activity patterns in specific neuronal ensembles as a mechanism for sleep-dependent facilitation of memory storage. While the phenomenon of replay during sleep has been widely reported[5,19,24,35,51], a causal role for sleep-dependent replay in memory consolidation has been difficult to prove. At least two technical obstacles have slowed progress toward understanding the role of replay in sleep-dependent consolidation. First, many tasks used in rodents to study phenomena (e.g., maze running) that require several days of training prior to obtaining recordings of sequential firing patterns—a timescale incompatible with memory consolidation occurring across a single sleep period. Second, many prior studies aimed at addressing the question of replay's necessity for consolidation have relied on disrupting circuit-level activity across windows of sleep[33,34], sometimes over several days[31,32]. Here we have taken advantage of recently developed genetic tools to label cue-activated neurons[38] and a new single trial paradigm for studying sleep-dependent consolidation of memory for a specific sensory cue (Fig. 1). These have allowed us to demonstrate that sleep-associated reactivation of cue-activated visual engram neurons plays a critical, instructive role in consolidating an associative memory linked to that cue.

Recent work has aimed at understanding the role of neuronal activity in, and communication between, subcortical and neocortical brain areas for promoting systems memory consolidation—which is essential for long-term storage of polymodal information in the brain. Behavioral findings suggest that sleep may play a vital role in this process. For example, in human subjects following training on tasks with implicit rules, subsequent sleep benefits gist extraction, abstraction, and correct rule application[3,52,53]. In other words, sleep appears to benefit the formation of new cognitive schemas. Our current behavioral findings support this idea and suggest a potential underlying mechanism. We find that, in mice that are SD, not only do mice not discriminate between cures, but overall freezing (i.e., fear response) is low (Fig. 1 and Extended Data Fig. S1). In contrast, in mice with sleep-targeted optogenetic inhibition of V1 engram neurons, mice do not discriminate between cues, and overall freezing is high (Fig. 7 and Extended Data Fig. S7). One interpretation is that, after sleep, fearful aspects of the memory (i.e., gist) are intact, but without activation of cue-selective neurons in V1, fear memory lacks the appropriate

specificity to the shock-paired visual cue. Together, this suggests that sleep-dependent reactivation of sensory engram neurons in neocortex may be a general feature of higher forms of associative learning that have specific sensory features.

A limitation of the present study is that inhibition of visual engram neurons in V1 occurred throughout all stages of sleep (i.e., both REM and NREM). Our prior work on experience-dependent plasticity in V1 has demonstrated that thalamocortical oscillations coordinating activity between the lateral geniculate nucleus (LGN) and V1 during NREM sleep are essential for orientation preference shifts in V1[11]. The pattern of optogenetic stimulation used on visual cue-activated neurons in this study (i.e., regular periodic activation at 1 Hz) is in some ways similar to what occurs in V1 during these NREM oscillations. Critically, this pattern of activation is sufficient to drive large V1 orientation preference shifts (Fig. 5). However, a role for REM activity in cortical plasticity cannot be ruled out. REM plays a critical role in developmentally regulated experience-dependent plasticity in V1[54]. In many species, REM is characterized by selective activation of LGN–V1 circuitry during pontine–geniculate–occipital waves, which promote synaptic plasticity in various brain structures[9]. Future work will be aimed at both characterizing patterns of activity in orientation-selective populations during REM vs. NREM and in targeting inhibition of this population to specific states.

The present findings converge with recent work in the olfactory system, where sensory representations of specific odorants are enhanced by odorant-associated aversive learning[40,55]. The data presented here may further inform our understanding of how sensory cortical areas interact with structures such as the hippocampus and amygdala during sleep and how these interactions inform consolidation of specific emotionally salient memories. Together our data indicate that primary sensory structures engaged in fear memory encoding communicate with structures conveying emotional valence information during post-learning sleep to promote long-lasting fear association with a specific cue. Whether this inter-regional communication is unique to one or more sleep states is a critical unanswered question. Answering this question may have important implications not only for understanding sleep's mechanistic role in memory consolidation but also its mechanistic role in regulation of mood and affect. It will also have specific implications for treating disorders where fear is dysregulated or misattributed, including anxiety and panic disorders, acute stress disorder, and posttraumatic stress disorder.

## Methods

**Animal handling and husbandry**. All animal procedures were approved by the University of Michigan Institutional Animal Care and Use Committee. With the exception of constant dark following tamoxifen administration, mice were kept on a 12-h:12-h light:dark (LD) cycle and were given food and water ad lib throughout the entirety of the study. Following surgical procedures, and during habituation prior to cued conditioning, mice were individually housed in standard caging with beneficial environmental enrichment (nesting material, manipulanda, and/or novel foods). Housing was maintained at 18–23 °C and 40–60% relative humidity throughout all experiments.

**Visually cued fear conditioning**. For 3 days prior to conditioning, mice were habituated to 5 min/day of gentle handling. Following the habituation period, at ZT0, mice underwent visually cued fear conditioning in a novel arena (context A). They were allowed 2 min to acclimate to the arena. They then experienced 3 pairings of a 30-s visual stimulus (presented simultaneously on 4 LED monitors surrounding the arena) co-terminating with a 2-s 0.75 mA foot shock. These pairings were divided with a 60-s intertrial interval. Each visual stimulus consisted of a 1-Hz phase-reversing oriented grating ($X°$) with a spatial frequency of 0.05 cycles/degree and contrast of 100%.

Following conditioning, C57BL/6J mice (Jackson) used for experiments outlined in Fig. 1 were returned to their home cage and were either allowed 12 h ad lib sleep or were SD using gentle handling (i.e., cage tapping, nest disturbance, and light touch with a cotton-tipped applicator to cause arousal from sleep) for 6 h, after which they were allowed 6 h ad lib sleep. All transgenic mice (see below) with

data shown in Figs. 2, 3, and 7 were allowed ad lib sleep in their home cage following conditioning.

At ZT12 (i.e., 12 h following conditioning), mice were placed in a dissimilar novel context B for cued fear memory testing. Context B differed from context A (used during conditioning) in several ways. The two arenas differed in size and shape (one a 18"×18"×18" open top arena, the other an 11.6"×9.25"×8.25" Med Associates conditioning chamber), odor, floor texture, and lighting condition.

Parameters for testing of fear responses were determined in a series of pilot studies, which are summarized in Extended Data Fig. S3. During testing, mice were exposed to two distinct oriented grating stimuli ($X°$ and $Y°$) to assess cue discrimination. At the start of each test, mice were allowed 3 min to acclimate to the arena, after which they were presented first with an oriented grating distinct from the shock cue ($Y°$) and later with a grating identical to the shock cue ($X°$), consistent with experiments assessing discrimination in auditory cued fear[56] and contextual fear[57]. Gratings were presented for 3 min followed by 1 min of post stimulus arena exploration. A minimum of 30 min separated the two grating presentations. Because pilot data indicated that discrimination was not contingent upon the orientations selected for shock and neutral cues (Extended Data Fig. S3c–f), for all subsequent tests, either 135° or 90° were used as the shock cue, and 0° was used consistently as the neutral cues.

Freezing responses were quantified for each grating stimulus using previously established criteria[58]. For each test, two scorers blinded to behavioral condition quantified periods of immobility during presentation of grating stimuli that included fear features, such as hyperventilation and rigid posture. Freezing during presentation of the two gratings was compared to calculate a discrimination index: (percentage of freezing during shock [$X°$] stimulus)/(percentage of freezing during shock [$X°$] stimulus + percentage of freezing during neutral [$Y°$] stimulus).

To test for the time-of-day effects on visually cued fear memory recall (Extended Data Fig. S3), additional cohorts of mice were trained at ZT0 as described above and tested at 12, 24, or 36 h later.

**Genetic tagging of orientation-selective V1 neurons**. Prior to all procedures for targeted recombination in visual engram neurons, mice were habituated for 3 days to gentle handling procedures. After habituation, at ZT0, the mice were placed in this square arena surrounded by 4 LED monitors. Each monitor presented a single orientation ($X°$) phase-reversing grating stimulus (1 Hz, 0.05 cycles/degree, 100% contrast) for 30 min (or, for negative controls shown in Fig. 2, a dark screen). Immediately after stimulus or dark screen presentation, mice received an intraperitoneal injection of tamoxifen (100 mg/kg in 95% corn oil/5% ethanol) and were placed in complete darkness for the next 3 days to prevent further visually driven recombination in V1. The timing of tamoxifen administration was aimed at targeting the known peak expression of endogenous *cfos* following visual stimulus presentation[59–61]. Following 3 days of constant dark housing, mice were returned to a normal 12 h:12 h LD cycle for 7 days prior to further experiments. cfos-CRE[ER] mice (Guenthner et al.; B6.129(Cg)-*Fos^{tm1.1(cre/ERT2)Luo}*/J; Jackson) crossed to B6.Cg-*Gt(ROSA)26Sor^{tm9(CAG-tdTomato)Hze}*/J, B6.Cg-*Gt(ROSA)26Sor^{tm32(CAG-COP4*H134R/EYFP)Hze}*/J, or B6.Cg-*Gt(ROSA)26Sor^{tm40.1(CAG-aop3/EGFP)Hze}*/J (Jackson) mice to induce CRE recombinase-mediated expression of tdTomato, ChR2, or ArchT. Controls for optogenetic experiments (characterizing behavioral and neuronal firing responses to light delivery alone) were done using mice that were handled and housed identically to those described above. These mice received injections of vehicle (corn oil) without tamoxifen following stimulus presentation.

**Histology and immunohistochemistry**. At the conclusion of each experiment, mice were deeply anesthetized with pentobarbital and transcardially perfused with saline and 4% paraformaldehyde. Brains were dissected, post-fixed, cryoprotected in 30% sucrose, and cryosectioned at 50 μm. Transgene expression in V1 was verified for all experiments using CRE-dependent transgenic lines prior to subsequent data analysis. For electrophysiological recordings in V1, electrode placement was verified prior to data analysis. Immunohistochemistry for cFos was carried out using rabbit-anti-cfos 1:1000 (Abcam; ab190289) and secondary donkey-anti-rabbit conjugated to Alexa Fluor 405 (1:200; Abcam; ab175651); coronal sections containing V1 were mounted using Fluoromount-G (Southern Biotech). Image acquisition was carried out using a Leica TCS SP5 confocal microscope, using a ×20 objective to obtain images spanning the layers of V1. The images were obtained as z stacks using 10 μm steps. Identical acquisition settings (e.g., exposure times) were used for all images taken from mice in all groups for experiments presented in Fig. 2 and separately for all images taken from mice in all groups for experiments presented in Fig. 4. tdTomato+ and cFos-immunopositive cell bodies were quantified in 5–6 sections of V1 from each mouse, by a scorer blinded to animal condition, using previously established procedures[62]. Co-labeling of tdTomato and anti-cFos was quantified using the Image J JACoP plugin (FIJI—ImageJ 1.52P). Co-labeling values for each mouse (averaged across the 5–6 sections), and approximate density of tdTomato+ and cFos+ neurons, are reported in Figs. 2 and 4.

**V1 visual response recordings, optogenetic manipulations, and data analysis**. For anesthetized recordings of V1 neurons' visual responses and firing, mice were anesthetized using a combination of isoflurane (0.5–0.8%) and 1 mg/kg

chlorprothixene (Sigma). Data were acquired using a 64-channel Plexon Omniplex recording system, using previously described methods[11,35]. A 2-shank, linear silicon probe (250 μm spacing between shanks) with 25 μm inter-electrode spacing (16 electrodes/shank; Cambridge Neurotech) was slowly advanced into V1 until stable recordings (with consistent spike waveforms continuously present for at least 30 min prior to baseline recording [see below]) were obtained. Orientation tuning curves for recorded neurons were then generated as described previously[11], by presenting a series of 8 full-field phase-reversing oriented gratings (0, 22.5, 45, 67.5, 90, 112.5, 135, or 157.5 degrees from horizontal, 1 Hz, 0.05 cycles/degree, 100% contrast, 10 s duration) and a blank screen (to evaluate spontaneous activity) presented repeatedly (4–8 times each) in an interleaved manner.

For recordings during rhythmic optogenetic activation of $X°$-activated V1 neurons in ChR2-expressing mice (Fig. 5), tuning curves were generated: (1) at baseline (immediately following establishment of stable neuronal recordings), (2) after a 20–30-min period without optogenetic manipulation, and (3) after a 20–30-min period of 1 Hz optogenetic stimulation. Optogenetic stimulation consisted of blue light pulses (10 ms, 473 nm, 10 mW power) delivered at 1 Hz. Only neurons stably recorded throughout all phases of the experiment (shown in Fig. 5a) were included in firing and visual response analysis.

To assess the effects of optogenetic inhibition of $X°$-activated V1 neurons in ArchT-expressing mice (Fig. 6), recordings consisted of a 30-min spontaneous activity recording with no manipulation, a 30-min recording with periodic inhibition (532 nm green light, 15 mW, delivered in cycles of 5 s on, followed by a 500-ms offramp and a 1-s off period). Following these recordings, two orientation tuning curves were generated for all recorded neurons: (1) a baseline without inhibition and (2) with inhibition of $X°$-activated V1 neurons occurring during 10-s presentations of oriented grating stimuli. Only neurons stably recorded throughout all phases of the experiment (shown in Fig. 6a) were included in firing and visual response analysis.

For all recordings, stable single units were isolated using principal component analysis-based analysis and multivariate analysis of variance-based cluster separation, implemented using the Offline Sorter software (version 4; Plexon) and previously described methods[11,35]. Units that could not be reliably discriminated or had refractory period violations in their spiking patterns were eliminated from subsequent analyses. Changes in orientation tuning were assessed relative to the orientation of the TRAPed ensemble ($X°$), based on neuron mean firing rate responses to gratings of different orientations. For each tuning curve, an orientation preference index (OPI) was calculated for $X°$ and the orthogonal stimulus orientation ($X°/X + 90°$), as described previously[11,42,43]. Percentage of changes in OPI (across optogenetic stimulation or control conditions) were calculated as $[(OPI^{pre} − OPI^{post})/OPI^{pre}] \times 100$. Firing responses of neurons during rhythmic optogenetic stimulation in ChR2-expressing mice was assessed from $Z$-scored perievent rasters centered on blue light onset; significance of time-locked excitation or inhibition was calculated based on positive or negative Z-score deviations beyond the 99% confidence interval (NeuroExplorer version 4; Plexon). Changes in firing during optogenetic inhibition in ArchT-expressing mice were calculated for each neuron within the inhibition recording period, by comparing mean firing rate during the last 1.5 s of each green light delivery period with mean firing rate during the subsequent 500-ms offramps and 1-s off period.

Power spectral density for LFPs was detrended using the NeuroExplorer software (version 4; Plexon) with a single taper Hann Windowing Function with 50% window overlap. These were averaged across all active electrodes on each silicon probe shank. Distributions of power (between 0 and 20 Hz) were compared statistically using K-S tests.

**Surgical procedures**. For V1 optical fiber implantation, mice were anesthetized using 1–2% isoflurane. Optical fibers (0.5 NA, 300 μm core, ThorLabs) were positioned bilaterally at the surface of V1 at a 80 degree angle relative to the cortical surface (2.9 mm posterior, 2.7 mm lateral). Implants were secured to the skull with an anchor screw positioned anterior to bregma, using Loctite adhesive. For EEG/ electromyography (EMG) recordings to differentiate sleep states, in addition to bilateral V1 optical fibers, mice received an EEG screw over V1 (2.9 mm posterior, 2.3 mm lateral), a reference screw over the cerebellum, and an additional EMG electrode in the nuchal muscle. Mice were allowed 10 days of postoperative recovery before procedures to induce transgene expression in V1.

**Optogenetic manipulations in behaving animals**. Two cohorts of implanted mice, expressing ChR2 in the TRAPed ensemble, were used to test perception of optogenetic activation of this cell population. Prior to behavioral training and testing, these mice were habituated to handling and tethering (for light delivery to V1) procedures for 3 days. The first cohort (Fig. 3b, c) underwent cued fear conditioning as described above in context A at ZT0, with 30-s blocks of rhythmic light delivery to V1 (1 Hz, 10 mW, 10 ms pulses) serving as a proxy shock cue (i.e., substituting for visual oriented grating presentation). Following 3 optogenetic stimulation-shock pairings (of equal duration and relative timing to visual stimulus presentation during training, as described above), these mice were returned to their home cages and allowed ad lib sleep until ZT12. At ZT12, they were placed in context B and freezing responses were assessed for visual presentation of both the same orientation as the TRAPed ensemble ($X°$) and an alternate orientation ($Y°$), as described above. A second cohort of mice (Fig. 3d–e) underwent visually cued fear

conditioning to the same angle as the TRAPed ensemble ($X°$) in context A at ZT0. After conditioning, they were returned to their home cage for ad lib sleep. At ZT12, they were tested in context B, where freezing behavior was assessed separately before, during, and after a period of 1 Hz light delivery to V1 (with percentage of freezing calculated for the 3 min before, 3 min during, and 1 min after optogenetic stimulation, immediately after which the mouse was removed from context B).

To assess the effects of sleep-targeted inhibition of visual engram neurons, 10 days after EEG/EMG and optical fiber implantation, mice underwent procedures to induce the expression of ArchT in the TRAPed orientation-specific ensemble. Following 3 days of habituation to handling and tethering (for light delivery to V1 and EEG/EMG recording), these mice underwent 12 h sleep/wake baseline recordings, starting at ZT0. The next day, mice underwent visually cued fear conditioning at ZT0, using either the same orientation as the TRAPed ensemble ($X°$) or an alternate orientation ($Y°$) as a cue for foot shock. They were then returned to their home cage for ad lib sleep. For the first 6 h post-conditioning, a subset of mice expressing ArchT underwent periodic optogenetic inhibition targeted to both NREM and REM sleep. The state targeting was based on EEG signals, EMG signals, and the animal's behavior. A control group of mice that were not expressing ArchT underwent the same light delivery and recording procedures. At ZT12, all mice were placed in context B to assess freezing responses to both $X°$ and $Y°$ oriented gratings, as described above.

EEG and EMG signals were used offline to classify each 10-s interval of baseline and post-conditioning recording periods as wake, NREM, or REM sleep, using the custom MATLAB software[11,35]. Additionally microarousals (periods of non-oscillatory activity between periods of NREM) as small as 5 s were identified as wake. Mean power spectral density was calculated separately within REM, NREM, and wake for each phase of recording and within and outside of periods of light delivery to V1 as described previously[11]. The power spectra were calculated as percentage of the total spectral power.

**Statistical methods**. All statistical analyses were done using GraphPad Prism (version 8) and SigmaPlot (version 14.0). Prior to making comparisons across values, the normality of distributions was tested using the D'Agostino–Pearson omnibus k2 test. Nonparametric tests were used when data distributions were non-normal or when $n$ values were too low to test normality. If the data involved multiple measurements from one animal (e.g., multiple images taken from the same animal for immunohistochemistry or multiple neurons recorded from individual mice), nested statistics were used. All statistical tests were two tailed. For each specific data set, the statistical tests used are listed in the "Results" section. $p$ values are represented as $*p < 0.05$, $**p \leq 0.01$, $***p \leq 0.001$, $****p \leq 0.0001$, respectively.

## Data availability

All relevant raw data and analysis tools are available upon reasonable request from the authors. Source data are provided with this paper.

## Code availability

MATLAB codes used for sleep scoring are available from the authors upon reasonable request.

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

## Acknowledgements

The authors are grateful to members of the Aton laboratory and to Dr. Natalie Tronson, Dr. Monica Dus, Dr. Richard Hume, and Dr. Dawen Cai for helpful feedback on this manuscript. This work was supported by research grants from the NIH (R01 NS104776) and the Human Frontiers Science Program (N023241-00_RG105) to S.J.A., a NSF Graduate Research Fellowship to B.C.C., and a Rackham Graduate Fellowship to B.C.C.

## Author contributions

B.C.C. and S.J.A. designed the experiments. B.C.C. led experimental implementation and data analysis with assistance from E.J.P., A.E., L.G., J.S., K.S., J.G.-A., M.Z., A.K.Y., F.R.K., J.D.M., L.W., S.J., and S.J.A. B.C.C. and S.J.A. wrote the manuscript.

## Competing interests

The authors declare no competing interests.
