## [Peer Review File · Nature Communications]

REVIEWER COMMENTS

Reviewer #1 (Remarks to the Author):

In this manuscript, Clawson et al. utilized a visually cued fear conditioning paradigm to test the importance of sleep and neuronal reactivation for fear memory consolidation. The authors first established that sleep deprivation disrupts fear memory recall compared with no sleep manipulation. They then established the specificity of the TRAP tagging system by tagging orientation-selective V1 neurons. Using this TRAP system, they tagged cFos-expressing neurons in V1 with ChR2 during visual stimulus presentation of one orientation (X). They fear conditioned (FC) mice with stimulation of the X-tagged population acting as the conditioned stimulus and footshock as the unconditioned stimulus. The authors separately demonstrated that stimulation of the tagged population after visually-cued FC was capable of driving freezing during a recall test. The X-tagged population was highly reactive (cFos+ during tagging and after FC) in the hours following FC using the same visual stimulus orientation (X) and not a different orientation (Y). In anesthetized X-tagged mice, stimulation of the tagged population led to an enhanced response to the tagged orientation, specifically for neurons that were activated by optogenetic stimulation. Conversely, inhibition of the same population during stimulus presentation led to a decrease in orientation preference, specifically for neurons inhibited by optogenetic inhibition. Last, the authors inhibited X-tagged V1 neurons during non-REM and REM sleep following FC, which led to fear generalization across X and Y orientations, but only if X was the conditioned stimulus. The reported results are very interesting and represent an advancement in our understanding of how sensory neuronal populations are modified during memory consolidation to support memory retrieval. The experimental designs are thoughtful, the combination of tools impressive, and these results are convergent with results from a notable study demonstrating enhanced sensory representations following olfactory fear conditioning (Dias et al, 2014), as well as a similar study that creates an olfactory fear conditioning memory in mice with just neuronal manipulation (Vetere et al., 2019). Overall, I am enthusiastic about the publication of these findings and believe they will be of wide interest to the field of neuroscience, particularly in the area of learning and memory. There are, however, several minor concerns which are discussed below.

Minor Points

- Were the X and Y orientations counterbalanced as the conditioning orientation throughout the study?
- Were there order effects of fear memory recall? That is, did mice freeze differently if X was presented before Y or after?
- Figure 3d-e: What were the time lengths of the pre/stim/post periods? Were they equal in length? Lines 367-374 describe recall experiments with cue presentation; are these times the same for the optogenetic experiment?
- Figure 4c is never referred to in the text.
- Figure 4b-c: Are the overall cFos expression levels the same across the X and Y groups? Please report those data. It is possible that since the mice are experiencing the X stimulus twice, there is less cFos induction in that group since the stimulus is less novel.
- Lines 206-207: What percentage of neurons does the TRAP system tag? Is this similar to the 4% of immediately active neurons following optogenetic stimulation?
- Figure 5: Stability of orientation-selectivity is discussed in lines 216-218; however, these results are not demonstrated in the figure. Please graphically represent these results.

- Figure 5e-f: The authors may consider re-labeling the x-axis "Pre/Post" rather than "no stim/stim" because the current labels give the impression that the stimulation period is being analyzed when it is actually the post-stimulus orientation assessment.

- Lines 219-221: The authors claim the shift in orientation preference was greater for the neurons activated by light than non-activated neurons; however, Figure 5e-f are not directly statistically compared. Rather, there is a selective effect in light-activated neurons with no effect in the non-activated neurons. This should be reworded, or a direct statistical test made between the two neuronal populations.

- Lines 224-225: If the shift is similar to that seen in V1 after stimulus presentation + sleep, why are all those cells not tagged with TRAP and all show the immediate light-induced response?

- Figure 5c and 6c: A frequency distribution of response profile changes (-100% to +100%) would depict the response profiles more richly (continuous metric) than is currently depicted with the pie charts (ordinal metric). Although 5e-f, 6e-f use subpopulations from the pies, it would be useful to see the whole population before neurons are pooled into defined categories.

o Figure 6f: Is this result driven by one of the 3 inhibited populations? If not, why are they broken up into 3 populations?

- If possible, please include no-opsin controls for optogenetic experiments. Since light stimulation can act as a visual stimulus, this could confound results – especially for experiments where behavioral responses are read out during light-on periods.

- Labeling the figure numbers would make it easier to refer to from the text.

I sign off on this review, Denise Cai

Reviewer #2 (Remarks to the Author):

The authors examine in mice the effects of sleep on visually-cued fear conditioning. First, they provide behavioural evidence that consolidation of such memory is sleep dependent. Second, they show that the fear response can be reinstated in a different context by reactivating cue responsive V1 cell ensembles. Third, they show that inhibiting cue specific cell ensembles during sleep leads to an unspecific fear response. The results confirm and substantially add to previous work showing that reactivation during sleep is essential for memory consolidation. Especially their finding that inhibition of V1 ensembles leads to a stronger but non-discriminative fear response has important implications on sleep's function for memory abstraction. This is overall highly impressive data. However, the authors' conceptual embedding and interpretation of findings is too much focussed on how sleep-consolidation strengthens memory and neglects that sleep-consolidation, at the same time, affects generalization and memory abstraction.

While in general I am very positive about the manuscript, I see several points that need thorough revision:

Major points:

1) As mentioned above my main concern with the manuscript pertains to the conceptual embedding and interpretation of the results. The abstract states "Optogenetic inhibition of the engram population during post-conditioning sleep disrupts consolidation of fear memory" (Similar statements are made in the introduction and discussion sections). However, the experiments show that the fear response is stronger and more generalized following inhibition of V1 engram cells compared to the fear response of control animals. At a glance, this behavioral finding indicates a stronger fear memory after inhibition of V1 engram cell during sleep (compared with the control), and thus apparently contradicts the authors' overall interpretation. Yet, I think these results very much support the concept that

consolidation during sleep follows a systems consolidation process in which replay of newly encoded memories reorganize memory representations such that representation of certain (gist) aspects become enhanced, in this case the differential V1 representation of the cue vs non-cue gratings. The concept of replay during sleep affecting systems consolidation (rather than only synaptic consolidation) should be adopted to appropriately frame the current findings (see e.g. Klinzing et al. Nat Neurosci 2019, for a recent update).

2) Related to my previous point it would be interesting to see if the fear response of those animals is also elicited without being exposed to any grating. This analysis would show whether the fear response becomes independent of the grating orientation and even occurs independently of any of the two (relatively similar) visual cues. To discriminate in addition possible contextual contributions, the authors might provide more detailed descriptions of the similarities and differences of the two contexts used for conditioning and recall. Were distal cues similar?

Further more minor points:

1) Several methodological issues need to be clarified: The authors provide additional data of visually-cued fear memory after 12h, 24h and 36h and the authors write that this was done in an "additional cohort of mice". However, the data shown in supplemental figure 2 is identical with the data shown for the male animals in figure 1b. Why did the authors ignore the females here? This is important because the SD group of this subsample fails to reach a significant difference for the discrimination index.

2) Related to this point the authors write: "To compare discrimination between cues, a discrimination index was calculated. Both freely sleeping and SD mice showed discrimination that differed from chance values, however this effect was far clearer in mice allowed ad lib sleep (Wilcoxon signed rank test; Sleep: $p = 0.0003$, SD: $p = 0.049$)". Please provide statistics for a direct comparison between sleep and SD animals instead of comparing p-values.

3) The data shown in fig. 5 and 6 panel e and f appears not to be normally distributed. Why did the authors choose nested t-test here? The same cells were tested under both conditions. Therefore, one would expect a test for repeated measures instead. Additionally, colour coding the data from the different animals (or showing the data separately) would help to exclude the possibility that the effect is mainly driven by data from one or two animals.

4) The authors write "visual behaviour" (figure 3). This term is confusing and should be replaced by a precise description in the figure legend and/or a small schematic in the figure itself. How are the intervals "pre", "stim" and "post" shown in figure 3 defined?

5) The assessment of how rhythmic offline reactivation and inhibition of the XO-activated TRAPed population affect firing in surrounding V1 neurons (figs 5 and 6) was done in anesthetized conditions which may differ from sleep. This should be acknowledged when discussing respective data (e.g., second to last paragraph of discussion).

Reviewer #3 (Remarks to the Author):

In this study, Clawson et al., test the hypothesis that cue-sensitive V1 neurons are critical for sleep-dependent memory consolidation. To address this question, they used TRAP to label and manipulate V1 neurons that respond to a visual CS and tested their behavioral function during fear learning and memory. They find that optogenetic activation of cue-sensitive neurons can serve as a CS during conditioning and can drive freezing during recall. In naïve animals, activation or inhibition of an orientation selective ensemble can induce plasticity in neighboring neurons, biasing their orientation preference relative to the TRAPed population. Finally, they show that inactivating cue-sensitive V1 neurons during post-conditioning sleep impairs behavioral discrimination during memory recall.

Overall, I think the authors use elegant methods to address an interesting, previously unanswered question. Previous work demonstrated that sequences of activity are replayed in V1 during sleep (Ji & Wilson 2006). In addition, it had been shown that optogenetic activation of ensembles of neurons in sensory cortex can act as a CS (Choi et al 2011). Here the authors go a step further, demonstrating that orientation-sensitive neurons can serve as a CS during fear conditioning, can drive cortical

plasticity after learning, and are necessary for post-sleep memory consolidation. The most interesting experiment is in Figure 7, where they show that silencing TRAPed V1 neurons during post-learning sleep impairs fear discrimination. However, more controls are needed to drive their main points home.

Major comments

Most of my concerns are related to their claim that 'cue-responsive neurons are selectively reactivated in V1 after post-conditioning sleep'. This conclusion is based on Figures 4–6, which I think need better controls. See below for details.

Figure 4C (and 2E) – The observed reduction in TRAP/Fos overlap could reflect the fact that the different populations of V1 neurons are activated by each orientation (one TRAPed and one Fos+ later), as the authors suggest. Alternatively, it's possible to see differences in overlap due to differences in the raw numbers of TRAPed and Fos+ neurons, which impacts the chance of overlap. The authors should report the numbers of TRAPed and Fos+ neurons and normalize to chance for their quantification of Fos/TRAP overlap. They should also report how they imaged the brain sections.

In Figure 5 and 6, the authors show that optogenetic reactivation or inhibition of X° TRAPed neurons induces plasticity such that more (Figure 5) / fewer V1 (Figure 6) neurons exhibit a preference for the X° orientation. Adding a GFP-only cohort of mice would control for effects of the 30-minute laser stimulation.

In addition, in Figure 3, they show that optogenetic stimulation of X°-TRAPed neurons can act as a CS during conditioning, or can elicit freezing following conditioning in which X° was the CS. To prove that these effects are indeed specific to X° population, they should do additional controls. For instance, they could trap Y° and test whether stimulating those neurons elicits freezing after conditioning with X° as the CS as in 3D. Alternatively, they could TRAP X° neurons with a GFP control virus instead of ChR2. In each case, the prediction would be that these optogenetic manipulations would not elicit higher levels of freezing.

Minor comments

The introduction is very short and the majority of it is a summary of this paper. To help put the study in context, it would be nice to discuss preceding work in more detail.

There is a typo in the introduction (line 69) "the mice behaviorally discrimination" should be "discriminated".

They should define 'ZT' at the beginning of the results section.

To label activated neurons, the authors injected tamoxifen immediately after the visual stimulation. In Guenthner et al., they show that you can achieve much higher numbers of labeled cells in V1 if you inject tamoxifen 24hrs before the stimulus. Why did they choose this TRAPing protocol?

Response to reviewer comments

Reviewer #1 (Remarks to the Author):

In this manuscript, Clawson et al. utilized a visually cued fear conditioning paradigm to test the importance of sleep and neuronal reactivation for fear memory consolidation. The authors first established that sleep deprivation disrupts fear memory recall compared with no sleep manipulation. They then established the specificity of the TRAP tagging system by tagging orientation-selective V1 neurons. Using this TRAP system, they tagged cFos-expressing neurons in V1 with ChR2 during visual stimulus presentation of one orientation (X). They fear conditioned (FC) mice with stimulation of the X-tagged population acting as the conditioned stimulus and footshock as the unconditioned stimulus. The authors separately demonstrated that stimulation of the tagged population after visually-cued FC was capable of driving freezing during a recall test. The X-tagged population was highly reactive (cFos+ during tagging and after FC) in the hours following FC using the same visual stimulus orientation (X) and not a different orientation (Y). In anesthetized X-tagged mice, stimulation of the tagged population led to an enhanced response to the tagged orientation, specifically for neurons that were activated by optogenetic stimulation. Conversely, inhibition of the same population during stimulus presentation led to a decrease in orientation preference, specifically for neurons inhibited by optogenetic inhibition. Last, the authors inhibited X-tagged V1 neurons during non-REM and REM sleep following FC, which led to fear generalization across X and Y orientations, but only if X was the conditioned stimulus. The reported results are very interesting and represent an advancement in our understanding of how sensory neuronal populations are modified during memory consolidation to support memory retrieval. The experimental designs are thoughtful, the combination of tools impressive, and these results are convergent with results from a notable study demonstrating enhanced sensory representations following olfactory fear conditioning (Dias et al, 2014), as well as a similar study that creates an olfactory fear conditioning memory in mice with just neuronal manipulation (Vetere et al., 2019). Overall, I am enthusiastic about the publication of these findings and believe they will be of wide interest to the field of neuroscience, particularly in the area of learning and memory. There are, however, several minor concerns which are discussed below.

We thank the reviewer for their enthusiasm, their useful comments below, and their helpful commentary for contextualizing our findings. We now refer to these olfactory conditioning studies in both our results section (where our findings converge with olfactory system findings) and in our revised discussion.

Minor Points

- Were the X and Y orientations counterbalanced as the conditioning orientation throughout the study?

We used a range of orientations as shock orientation X□ and neutral orientation Y□ in pilot studies for the cued conditioning. As shown in new Extended Data Figure S3, the choice of these orientations did not seem to affect discrimination performance. However, it is possible that differences in baseline orientation preference (which is not uniformly represented in V1, with more neurons preferring horizontal and vertical orientations) could lead to variability of outcome measures, even if presentations were random for each experiment. Thus for consistency across experimental groups, we used one of two different orientation combinations in the optogenetic manipulations. This is now described in the revised Materials and Methods section.

- Were there order effects of fear memory recall? That is, did mice freeze differently if X was presented before Y or after?

This is an excellent question, and yes, there were order effects. Thus we tested mice with visual cues in a manner similar to what has been reported for other forms of cued fear ¹. Our pilot studies (data shown in new Extended Data Figure S3) demonstrated that when X□ was presented before Y□ during fear memory tests, freezing was immediately higher than when Y□ was presented first. Consistent with results of contextual fear discrimination studies ², after immediate presentation of X□, freezing remained elevated (including during subsequent presentation of Y□). This suggests that fear responses during recall in response to presentation of X□ leads to heightened generalized fear throughout subsequent testing. For this reason, we presented the neutral stimulus Y□ prior to the shock stimulus X□ for our main experiments. For clarity, we now show these behavioral data in Extended Data Figure S3A-B, and discuss this issue in the revised Methods.

- Figure 3d-e: What were the time lengths of the pre/stim/post periods? Were they equal in length? Lines 367-374 describe recall experiments with cue presentation; are these times the same for the optogenetic experiment?

We are grateful to the reviewer for pointing out that the time courses were omitted from the text, for both Figure 3d-e and Figure 3b-c. For Figure 3d-e, the pre-stimulus, stimulus, and post-stimulus periods were 3 min, 3 min, and 1 min duration, respectively. We apologize for the lack of clarity in the previous text, which we have now modified (in the corresponding Materials and Methods subsection and Figure Legends) to provide these details for readers.

- Figure 4c is never referred to in the text.

We appreciate the reviewer pointing this out, and now include a callout to Figure 4c in the appropriate part of the revised Results section.

- Figure 4b-c: Are the overall cFos expression levels the same across the X and Y groups? Please report those data. It is possible that since the mice are experiencing the X stimulus twice, there is less cFos induction in that group since the stimulus is less novel.

This is an excellent question. There were no significant differences in cFos+ neuron density (or tdTomato+ neuron density) between the X° and Y° groups. This is true for both data presented in revised Figure 2 and Figure 4, which have both been amended to include this data. As now shown in those figures, we do not see differences between cFos expression following first X° and repeated X° presentation.

- Lines 206-207: What percentage of neurons does the TRAP system tag? Is this similar to the 4% of immediately active neurons following optogenetic stimulation?

This is a fantastic question, which has important implications for data interpretation in our study. We have quantified this, and now report this number, based on the known neuronal density in mouse V1 and the range of tdTomato labeling densities we measure following X° presentation (now shown in revised Figures 2 and 4), as 0.9-2.8%, which is very similar to the 4% of directly activated neurons we observe during optogenetic stimulation. This information is now included in the revised Results section. We thank the reviewer for suggesting this analysis.

- Figure 5: Stability of orientation-selectivity is discussed in lines 216-218; however, these results are not demonstrated in the figure. Please graphically represent these results.

We are grateful to the reviewer for pointing out that this wording doesn't match the data shown in the figure. We should have said, simply, that there is no increase in preference for the X° orientation in control conditions. We have reworded our discussion of these data in the Results and Discussion sections, to clarify that we see no increase in preference for X° in the no laser and the new no tamoxifen control + laser conditions.

- Figure 5e-f: The authors may consider re-labeling the x-axis "Pre/Post" rather than "no stim/stim" because the current labels give the impression that the stimulation period is being analyzed when it is actually the post-stimulus orientation assessment.

This is a great suggestion, and we have modified the axes of graphs in revised Figures 5 and 6 for clarity.

- Lines 219-221: The authors claim the shift in orientation preference was greater for the neurons activated by light than non-activated neurons; however, Figure 5e-f are not directly statistically compared. Rather, there is a selective effect in light-activated neurons with no effect in the non-activated neurons. This should be reworded, or a direct statistical test made between the two neuronal populations.

This is an excellent point, and we have reworded the discussion of these results for clarity. We thank the reviewer for pointing this out.

- Lines 224-225: If the shift is similar to that seen in V1 after stimulus presentation + sleep, why are all those cells not tagged with TRAP and all show the immediate light-induced response?

We apologize for failing to write this clearly in our initial submission. We intended to say that the shift is similar in nature (i.e., direction) – and much larger, on average, in magnitude – with respect to our prior findings after stimulus presentation + sleep. We have made that sentence in the text clearer to reflect our original intention. It is probably worth mentioning that some of the original work on visually-driven plasticity (which was published by the Bear lab) showed that the response changes induced are graded, with repeated stimulation (over several successive days) required to maximize the shift in favor of the stimulus orientation (Frenkel et al., 2006). For comparison, what we see after a single presentation + sleep is enough to significantly affect neurons' relative responses to $X^\circ/X+90^\circ$ (increasing this ratio ~20-25%), and only modest changes in the proportion of all V1 neurons having X° as their preferred stimulus orientation (~10-15% increases; Durkin et al., 2017).

- Figure 5c and 6c: A frequency distribution of changes (-100% to +100%) would depict the response profiles more richly (continuous metric) than is currently depicted with the pie charts (ordinal metric). Although 5e-f, 6e-f use subpopulations from the pies, it would be useful to see the whole population before neurons are pooled into defined categories.

We appreciate this comment, and completely agree. We now show firing histograms for ChR2-expressing mice (with data summarized in Figure 5) for all significantly activated neurons in Extended Data Figure S4. These graphs show the relative peak firing times for each neuron. We also now include exact firing rate changes for all neurons recorded from Arch-expressing mice in revised Figure 6. We thank the reviewer for providing this suggestion for improving the data presentation for these recordings.

o Figure 6f: Is this result driven by one of the 3 inhibited populations? If not, why are they broken up into 3 populations?

This is an excellent point, and we agree that there is no compelling reason to break the inhibited neurons into 3 populations in Figure 6. We had originally done this breakdown somewhat arbitrarily with an aim toward parallel aesthetic construction with Figure 5. Because there was no strong scientific premise underlying this decision, we now collapse inhibited neurons into one group for simplicity in the revised figure. We thank the reviewer for pointing out that this was making data interpretation unnecessarily complicated.

- If possible, please include no-opsin controls for optogenetic experiments. Since light stimulation can act as a visual stimulus, this could confound results – especially for experiments where behavioral responses are read out during light-on periods.

We thank the reviewer for this excellent suggestion. We agree that having no-opsin controls for these experiments will significantly improve the manuscript. We have now included data for mice of the same genotype, with sham induction of ChR2 or Arch expression. These mice were recorded under identical conditions to the mice shown in

Figures 5 and 6. These mice show no significant firing rate changes with light delivery, or consistent changes in orientation preference across the experiment.

- Labeling the figure numbers would make it easier to refer to from the text.

We apologize for making the manuscript more difficult to review than necessary – we now include legends with the figures in the manuscript document.

I sign off on this review, Denise Cai

Reviewer #2 (Remarks to the Author):

The authors examine in mice the effects of sleep on visually-cued fear conditioning. First, they provide behavioural evidence that consolidation of such memory is sleep dependent. Second, they show that the fear response can be reinstated in a different context by reactivating cue responsive V1 cell ensembles. Third, they show that inhibiting cue specific cell ensembles during sleep leads to an unspecific fear response. The results confirm and substantially add to previous work showing that reactivation during sleep is essential for memory consolidation. Especially their finding that inhibition of V1 ensembles leads to a stronger but non-discriminative fear response has important implications on sleep's function for memory abstraction. This is overall highly impressive data. However, the authors' conceptual embedding and interpretation of findings is too much focussed on how sleep-consolidation strengthens memory and neglects that sleep-consolidation, at the same time, affects generalization and memory abstraction.

While in general I am very positive about the manuscript, I see several points that need thorough revision:

We thank the reviewer for this insightful comment, and we agree that the issue of memory generalization/abstraction is an important one. We have revised the manuscript to hone in better on this point, as described in detail below.

Major points:

1) As mentioned above my main concern with the manuscript pertains to the conceptual embedding and interpretation of the results. The abstract states "Optogenetic inhibition of the engram population during post-conditioning sleep disrupts consolidation of fear memory" (Similar statements are made in the introduction and discussion sections). However, the experiments show that the fear response is stronger and more generalized following inhibition of V1 engram cells compared to the fear response of control animals. At a glance, this behavioral finding indicates a stronger fear memory after inhibition of V1 engram cell during sleep (compared with the control), and thus apparently contradicts the authors' overall interpretation. Yet, I think these results very much support the concept that consolidation during sleep follows

an systems consolidation process in which replay of newly encoded memories reorganize memory representations such that representation of certain (gist) aspects become enhanced, in this case the differential V1 representation of the cue vs non-cue gratings. The concept of replay during sleep affecting systems consolidation (rather than only synaptic consolidation) should be adopted to appropriately frame the current findings (see e.g. Klinzing et al. Nat Neurosci 2019, for a recent update).

We are very grateful to the reviewer for this insight. We agree with the interpretation that there appear to be two processes occurring during sleep. The first, which appears to be dependent on specific V1 ensembles, provides specific information to associate fear with the shock-paired stimulus (and not the neutral stimulus). The second, which is expressed in sleeping mice even when V1 ensembles are inhibited, enhances (generalized) fear responses overall. We speculate, as the reviewer does, that the increased overall fear we see in mice with sleep-targeted ensemble inhibition is due to a breakdown in systems consolidation – e.g. disrupted communication between visual system circuitry and circuits engaged in fear expression, such as the amygdala. We now provide a more in-depth discussion of our behavioral findings in the context of what is known regarding sleep’s role in systems consolidation. This discussion can now be found in the revised Discussion section of the manuscript. We appreciate the reviewer’s suggestion for this, and the recent reference. Inclusion of this idea strengthens the discussion of our present findings significantly.

2) Related to my previous point it would be interesting to see if the fear response of those animals is also elicited without being exposed to any grating. This analysis would show whether the fear response becomes independent of the grating orientation and even occurs independently of any of the two (relatively similar) visual cues. To discriminate in addition possible contextual contributions, the authors might provide more detailed descriptions of the similarities and differences of the two contexts used for conditioning and recall. Were distal cues similar?

We appreciate this suggestion, and have revised the manuscript to address it in a few ways. First, we now provide additional information regarding the similarities and differences of contexts A and B in the revised Materials and Methods section. The contexts differed in scent, floor texture, arena size and shape, and ambient lighting (the main distal cue difference between them). Second, we now provide measurements of freezing responses in both sleeping vs. sleep deprived mice, and inhibited vs. non-inhibited mice, as a function of time during testing – including the periods prior to cue presentation. These data show that there are low levels of freezing during the pre-cue phase under most conditions. Both sleep deprived mice and mice allowed *ad lib* sleep freeze significantly more to the shock cue than to the pre-cue blank screen (data now presented in Extended Data Figure S1). For the mice with optogenetic inhibition, both control groups showed significantly higher freezing to the shock cue than the pre-cue blank screens. The mice who had inhibition targeting neurons activated by the shock angle, generalized fear across angles alongside showing high levels of freezing to the pre-cue blank screen (data now presented in Extended Data Figure S7).

Further more minor points:

1) Several methodological issues need to be clarified: The authors provide additional data of visually-cued fear memory after 12h, 24h and 36h and the authors write that this was done in an “additional cohort of mice”. However, the data shown in supplemental figure 2 is identical with the data shown for the male animals in figure 1b. Why did the authors ignore the females here? This is important because the SD group of this subsample fails to reach a significant difference for the discrimination index.

We thank the reviewer for pointing out that we did not clearly label these data as replicated from Figure 1 and S1. Males only were used for this experiment to reduce overall mouse numbers, and because this supplemental experiment was of limited relevance to the primary focus of the paper. We have removed Extended Data Figure S1 in the revised manuscript for simplicity, as the issue of recall diurnality is tangential to (and outside the scope of) the main points of the paper.

2) Related to this point the authors write: “To compare discrimination between cues, a discrimination index was calculated. Both freely sleeping and SD mice showed discrimination that differed from chance values, however this effect was far clearer in mice allowed ad lib sleep (Wilcoxon signed rank test; Sleep: $p = 0.0003$, SD: $p = 0.049$)”. Please provide statistics for a direct comparison between sleep and SD animals instead of comparing p-values.

The direct comparison of the sleep and SD discrimination indices is *N.S.* (unpaired t test). The phenotype becomes clear, however, when examining the magnitude of freezing to the shock angle. The sleeping mice freeze significantly more to the shock cue than SD mice ($p = 0.007$, Holm-Sidak *post hoc* test). We provide additional support for this argument in new Extended Data Figure S1.

3) The data shown in fig. 5 and 6 panel e and f appears not to be normally distributed. Why did the authors choose nested t-test here? The same cells were tested under both conditions. Therefore, one would expect a test for repeated measures instead. Additionally, colour coding the data from the different animals (or showing the data separately) would help to exclude the possibility that the effect is mainly driven by data from one or two animals.

Here we used a nested t test to verify that multiple measures from neurons recorded from individual animals were not skewing our results. We have clarified this issue in the revised “Statistic Methods” subsection of the revised Materials and Methods section.

4) The authors write “visual behaviour” (figure 3). This term is confusing and should be replaced by a precise description in the figure legend and/or a small schematic in the figure itself. How are the intervals “pre”, “stim” and “post” shown in figure 3 defined?

We thank the reviewer for bringing this to our attention. We have removed this confusing term from the paper. We have clarified the timing of the “pre”, “stim” and “post” intervals in both the Figure 3 legend and in the revised Materials and Methods section.

5) The assessment of how rhythmic offline reactivation and inhibition of the XO-activated TRAPed population affect firing in surrounding V1 neurons (figs 5 and 6) was done in anesthetized conditions which may differ from sleep. This should be acknowledged when discussing respective data (e.g., second to last paragraph of discussion).

We agree completely that this is an important caveat. We have modified the text of the discussion accordingly, to reflect this, and thank the reviewer for this suggestion.

Reviewer #3 (Remarks to the Author):

In this study, Clawson et al., test the hypothesis that cue-sensitive V1 neurons are critical for sleep-dependent memory consolidation. To address this question, they used TRAP to label and manipulate V1 neurons that respond to a visual CS and tested their behavioral function during fear learning and memory. They find that optogenetic activation of cue-sensitive neurons can serve as a CS during conditioning and can drive freezing during recall. In naïve animals, activation or inhibition of an orientation selective ensemble can induce plasticity in neighboring neurons, biasing their orientation preference relative to the TRAPed population. Finally, they show that inactivating cue-sensitive V1 neurons during post-conditioning sleep impairs behavioral discrimination during memory recall.

Overall, I think the authors use elegant methods to address an interesting, previously unanswered question. Previous work demonstrated that sequences of activity are replayed in V1 during sleep (Ji & Wilson 2006). In addition, it had been shown that optogenetic activation of ensembles of neurons in sensory cortex can act as a CS (Choi et al 2011). Here the authors go a step further, demonstrating that orientation-sensitive neurons can serve as a CS during fear conditioning, can drive cortical plasticity after learning, and are necessary for post-sleep memory consolidation. The most interesting experiment is in Figure 7, where they show that silencing TRAPed V1 neurons during post-learning sleep impairs fear discrimination. However, more controls are needed to drive their main points home.

We thank the reviewer for their positive and thoughtful comments on our manuscript, which we address below.

Major comments

Most of my concerns are related to their claim that 'cue-responsive neurons are selectively reactivated in V1 after post-conditioning sleep'. This conclusion is based on Figures 4–6, which I think need better controls. See below for details.

Figure 4C (and 2E) – The observed reduction in TRAP/Fos overlap could reflect the fact that the different populations of V1 neurons are activated by each orientation (one TRAPed and one Fos+ later), as the authors suggest. Alternatively, it's possible to see differences in overlap due

to differences in the raw numbers of TRAPed and Fos+ neurons, which impacts the chance of overlap. The authors should report the numbers of TRAPed and Fos+ neurons and normalize to chance for their quantification of Fos/TRAP overlap. They should also report how they imaged the brain sections.

The reviewer makes an excellent point, and we now include data on the total numbers of tdTomato-expressing and cFos-expressing neurons per cubic mm for revised Figures 2 and 4. These data show that there significant differences between groups in the density of cFos+ or TRAPed cells in V1. We also now include more information on how brain sections were imaged, and how images were quantified, in the revised Materials and Methods section.

In Figure 5 and 6, the authors show that optogenetic reactivation or inhibition of X° TRAPed neurons induces plasticity such that more (Figure 5) / fewer V1 (Figure 6) neurons exhibit a preference for the X° orientation. Adding a GFP-only cohort of mice would control for effects of the 30-minute laser stimulation.

We thank the reviewer for this excellent suggestion. We agree that having no-opsin controls for these experiments will significantly improve the manuscript. We have now included data for mice of the same genotype, with sham induction of ChR2 or Arch expression. These mice were recorded under identical conditions to the mice shown in Figures 5 and 6. These mice show no significant firing rate changes with light delivery, or consistent changes in orientation preference across the experiment.

In addition, in Figure 3, they show that optogenetic stimulation of X°-TRAPed neurons can act as a CS during conditioning, or can elicit freezing following conditioning in which X° was the CS. To prove that these effects are indeed specific to X° population, they should do additional controls. For instance, they could trap Y° and test whether stimulating those neurons elicits freezing after conditioning with X° as the CS as in 3D. Alternatively, they could TRAP X° neurons with a GFP control virus instead of ChR2. In each case, the prediction would be that these optogenetic manipulations would not elicit higher levels of freezing.

We agree that having no-opsin controls for these experiments will significantly improve the manuscript. Similar to the point described above, we now include data for mice of the same genotype, with sham induction of ChR, which were conditioned and optogenetically “stimulated” identically to those in Figure 3. As we now show in Figure 3D, no-opsin controls do not show significant discrimination between cues. In Figure 3G we show that the no-opsin control mice do not freeze during optogenetic stimulation.

Minor comments

The introduction is very short and the majority of it is a summary of this paper. To help put the study in context, it would be nice to discuss preceding work in more detail.

We appreciate the reviewer’s point of view, and have revised the introduction accordingly. Specifically, we now discuss prior findings suggesting a specific role for

sleep in promoting memory consolidation for tasks involving emotional salience or implicit rules. We also discuss prior work showing a role for sleep in promoting systems memory consolidation, and in communication between neocortical and subcortical structures.

There is a typo in the introduction (line 69) “the mice behaviorally discrimination” should be “discriminated”.

We appreciate the reviewer pointing out this typographical error, which has been corrected.

They should define ‘ZT’ at the beginning of the results section.

We thank the reviewer for pointing this out, and have corrected this omission.

To label activated neurons, the authors injected tamoxifen immediately after the visual stimulation. In Guenther et al., they show that you can achieve much higher numbers of labeled cells in V1 if you inject tamoxifen 24hrs before the stimulus. Why did they choose this TRAPing protocol?

The reviewer’s point is quite valid – in the Guenther et al. paper, more V1 neurons were labeled by TRAP when tamoxifen was administered 24 h prior to presentation of a diffuse light stimulus (relative to time points at 0, 6, 12, or 36 h prior to stimulus). It is unclear to us why this was the case, as it is well documented that endogenous *cfos* mRNA and cFos protein expression peaks within 0.5 and ~1-2 h, respectively, following onset of a visual stimulus³⁻⁵. A plausible explanation is that in the Guenther study, 3 total days of prior visual deprivation led to homeostatic plasticity in the cortex, which may peak around that time⁶. Indeed, the diffuse stimuli used for testing in this study are ideal for testing homeostatic response changes in V1, and are frequently used in studies of homeostatic plasticity.

We wished to avoid the issue of homeostatic (and generally non-stimulus-specific) genetic tagging of neurons as potential confound. Our intention was not to label the maximal number of neurons, but to label those neurons which were most selectively responsive to the oriented grating presented. For this, we chose a time point which was most proximal to the stimulus itself (i.e., immediately following it), where visually-driven activity in the *cfos* promoter should be highest. As we now show in revised Figure 2, the number of cells labeled is relatively few compared to the number of cFos immunolabeled cells following repeat visual experience – suggesting that this is indeed the case. We have clarified our rationale for timing the tamoxifen the way we chose to do in the revised Materials and Methods section.

- 1 Antunes, R. & Moita, M. A. Discriminative Auditory Fear Learning Requires Both Tuned and Nontuned Auditory Pathways to the Amygdala. *J Neurosci* **30**, 9782-9787 (2010).
- 2 Wotjak, C. T. Sound check, stage design and screen plot – how to increase the comparability of fear conditioning and fear extinction experiments. *Psychopharmacology* **236**, 33-48 (2019).
- 3 Chaudhuri, A., Zangenehpour, S., Rahbar-Dehgan-F. & Ye, F. Molecular maps of neural activity and quiescence. *Acta Neurobiol Exp* **60**, 403-410 (2000).
- 4 Kaminska, B., Kaczmarek, L. & Chaudhuri, A. Visual Stimulation Regulates the Expression of Transcription Factors and Modulates the Composition of AP-1 in the Visual Cortex. *J Neurosci* **16**, 3968-3978 (1996).
- 5 Zangenehpour, S. & Chaudhuri, A. Differential induction and decay curves of c-fos and zif268 revealed through dual activity maps. *Molecular Brain Research* **109**, 221-225 (2002).
- 6 Lee, H.-K. & Kirkwood, A. Mechanisms of Homeostatic Synaptic Plasticity in vivo. *Front Cell Neurosci* **13** (2019).

REVIEWERS' COMMENTS

Reviewer #1 (Remarks to the Author):

The authors have addressed the comments and I recommend this study for publication.

Minor Points

- Extended Fig 3d-f: Please add y-axis labels for these three graphs.
- Extended Fig 3d-f: The line and circles connecting the two bars do not add information over the bars. However, the addition of individual datapoints for each mouse would provide valuable information about the distribution of freezing values. It would also make this figure consistent with graphs in the other figures where individual datapoints are represented.
- Lines 164-166; 183-185; lines 466-468: Were control mice still scruffed and injected with vehicle drug, or were they not handled at all? Scruffing and i.p. injection could act as a form of stress and could contribute to subsequent neuronal dynamics. Please update the methods with this detail.
- Line 503: How was "baseline" activity for each neuron calculated?

I sign off on this review- Denise Cai, Assistant Professor at Icahn School of Medicine at Mount Sinai

Reviewer #2 (Remarks to the Author):

The authors have satisfactorily addressed all of my points. I am very enthusiastic. The ms, in my view, is now excellent and ready to be published. Congratulations to authors!

Reviewer #3 (Remarks to the Author):

I think this is a very interesting and well-executed study. In my opinion, the authors sufficiently address the reviewer concerns.

I just have a few remaining minor concerns:

While I appreciate that they expanded the introduction, the focus on cortical-subcortical interactions in the new paragraph is a bit misleading given the sole focus on cortex in this paper. Perhaps tone that down a little bit.

Add axis labels to new graphs in Figure S3d-f.

They did not fully address my concerns about the TRAP/Fos overlap because they did not normalize to chance. Showing that there are no significant differences between numbers of Fos+ and TRAPed cells helps. However, the subsequent in vivo recordings and manipulations are more important in supporting their conclusions, so I am willing to accept the modifications that they made.

REVIEWERS' COMMENTS

Reviewer #1 (Remarks to the Author):

The authors have addressed the comments and I recommend this study for publication.

We are very grateful for the reviewer's thoughtful comments throughout the review process, which have significantly improved data presentation in our manuscript.

Minor Points

- Extended Fig 3d-f: Please add y-axis labels for these three graphs.

We thank the reviewer for bringing this omission to our attention. The y-axes are now labeled.

- Extended Fig 3d-f: The line and circles connecting the two bars do not add information over the bars. However, the addition of individual datapoints for each mouse would provide valuable information about the distribution of freezing values. It would also make this figure consistent with graphs in the other figures where individual datapoints are represented.

We thank the reviewer for letting us know that our data presentation was unclear. We have removed the lines as suggested, but have also clarified in the text that each of the panels shows data from only one mouse with different X and Y orientations selected for the shock and neutral cue.

- Lines 164-166; 183-185; lines 466-468: Were control mice still scruffed and injected with vehicle drug, or were they not handled at all? Scruffing and i.p. injection could act as a form of stress and could contribute to subsequent neuronal dynamics. Please update the methods with this detail.

We appreciate the reviewer bringing this to our attention – the handling of controls was not clearly described in the Materials and Methods section. These control mice received i.p. injections of corn oil vehicle, without tamoxifen, following presentation of visual stimuli. We have revised the description of our controls in both Materials and Methods and Results section.

- Line 503: How was “baseline” activity for each neuron calculated?

We have clarified this part of the Materials and Methods section, to indicate that these recordings occurred immediately following obtaining stable recordings. The text in this paragraph and the prior one have been modified accordingly.

I sign off on this review- Denise Cai, Assistant Professor at Icahn School of Medicine at Mount Sinai

Reviewer #2 (Remarks to the Author):

The authors have satisfactorily addressed all of my points. I am very enthusiastic. The ms, in my view, is now excellent and ready to be published. Congratulations to authors!

We are very grateful to the reviewer for their thoughtful comments throughout this process. Their suggestions have significantly improved the manuscript.

Reviewer #3 (Remarks to the Author):

I think this is a very interesting and well-executed study. In my opinion, the authors sufficiently address the reviewer concerns.

I just have a few remaining minor concerns:

While I appreciate that they expanded the introduction, the focus on cortical-subcortical interactions in the new paragraph is a bit misleading given the sole focus on cortex in this paper. Perhaps tone that down a little bit.

We appreciate the reviewer's suggestion; they raise an excellent point. We have added a final sentence to the opening paragraph of the introduction to narrow the scope of the questions being addressed in our specific study.

Add axis labels to new graphs in Figure S3d-f.

We thank the reviewer for bringing this omission to our attention. The y-axes are now labeled.

They did not fully address my concerns about the TRAP/Fos overlap because they did not normalize to chance. Showing that there are no significant differences between numbers of Fos+ and TRAPed cells helps. However, the subsequent in vivo recordings and manipulations are more important in supporting their conclusions, so I am willing to accept the modifications that they made.

We appreciate the reviewer's comments on this point, and we are glad that the data presented from in vivo recordings with optogenetic stimulation have helped to address this concern.